# Deciphering the genetic basis of grain iron and zinc content in wheat under heat and drought stress using GWAS

Sahana Police Patil[1], Hari Krishna[1,*], Narayana Bhat Devate[1],
Karthik Kumar Manjunath[1], P. N. Vinodh Kumar[1], Divya Chauhan[1], Shweta Singh[1],
Hanif Khan[2], Chandra Nath Mishra[2], Neelu Jain[1], Gyanendra Pratap Singh[3],
Pradeep Kumar Singh[1]*

1 Division of Genetics, ICAR-Indian Agricultural Research Institute, New Delhi, India, 2 ICAR-Indian
Institute of Wheat and Barley Research, Karnal, Haryana, India, 3 National Bureau of Plant Genetic
Resources, New Delhi, India

* harikrishna.agri@gmail.com (HK); pksinghiari@gmail.com (PKS)

pone.0329578

University, INDIA

**Peer Review History:** PLOS recognizes the
benefits of transparency in the peer review
process; therefore, we enable the publication
of all of the content of peer review and
author responses alongside final, published
articles. The editorial history of this article is
available here: https://doi.org/10.1371/journal.
pone.0329578

## Abstract

Wheat, a crucial food crop, is inherently deficient in essential micronutrients such
as iron and zinc. Climate change exacerbates its vulnerability to abiotic stresses
like drought and heat. Developing varieties that are both climate-resilient and
nutrient-dense offers a sustainable approach. The objective of this study was to iden-
tify genomic regions linked to grain Fe content (GFeC), grain Zn content (GZnC) and
thousand grain weight (TGW) traits in wheat grains subjected to heat and drought
stress through genome-wide association studies (GWAS). A genetically diverse set
of 280 wheat genotypes was assessed across three conditions: timely sown, late
sown (heat stress), and restricted irrigation (drought stress) over two years. Variation
in iron and zinc levels among genotypes was significant among the conditions, with
moderate heritability. Through GWAS 37 significant MTAs across the conditions were
identified. For thousand grain weight (TGW) 12 MTAs, for grain Fe content (GFeC) 14
MTAs, and for grain Zn content (GZnC) 11 MTAs were detected. Notably, four MTAs
for GFeC two of which were specific to heat stress were located on chromosome
7A. Among these, AX-94432820 (LSIR_23) resides near a RING-H2 finger protein
gene involved in metal-ion binding. Additionally, the stable SNP AX-94953068, also
on 7A, is adjacent to *TraesCS7A02G171600*, a gene implicated in stress response.
For GZnC, the stable SNP AX-95001849 ($r^2 = 12.89\%$) was significant under both
TSIR and TSRI, it maps to a plasma membrane ATPase. Using multivariate analysis,
MGIDI scores were calculated, identifying nine genotypes that excelled for all three
traits and conditions: RAJ4546, UP3063, HD3334, DBW296, MP1368, DBW333,
UP3058, DBW332, and BRW3863. These findings will support biofortification
breeding of the nutri-rich wheat varieties.

**Data availability statement:** All relevant data are within the manuscript and its Supporting Information files.

**Funding:** Part of the research was supported by a grant from the Bill & Melinda Gates Foundation (Grant 467 number # OPP1215722) sub-grant to India for the Zn mainstreaming project. The funders had no role in study design, data collection and analysis, decision to publish, or preparation of the manuscript.

**Competing interests:** The authors declare that the research was conducted without any commercial or financial relationships that could be construed as a potential conflict of interest.

## 1. Introduction

Wheat (*Triticum aestivum* L.) is one of the oldest among the cereals and critically important out of many recognized species. Wheat is the most widely consumed dietary grain and a significant source of protein, vital nutrients, and daily energy worldwide [1]. Since it is a major constituent of the diet around approximately 33% of people worldwide, it is crucial for ensuring sustaining global food availability and nutritional well-being [2]. Bread wheat have low micronutrient concentrations especially iron (Fe) and zinc (Zn), it is further reduced by milling [3,4]. Hidden hunger is mostly caused by a high-energy but low-nutrient diet, which is most common in developing nations like India [5]. Despite their small requirements, lack of essential micronutrients can contribute to illness and death [6]. Iron deficiency causes anemia and it affects around a quarter of the world's population [7]. Zinc deficiency affects around one in six individuals globally and is associated with various health disorders [8]. Zinc is a vital micronutrient involved in regulating growth and defense system function and is essential for the synthesis and activity of numerous enzymes [9]. Thousand Grain Weight (TGW) is a key agronomic trait contributing to final grain yield and market value in wheat. It is a complex, quantitative trait influenced by multiple genetic loci and significantly affected by environmental conditions, particularly heat and drought stress during the grain filling period. These abiotic stresses not only reduce grain number and size but also alter the physiological and metabolic pathways that influence assimilate partitioning and grain development [10]. Beyond yield, TGW is increasingly recognized for its relationship with grain nutritional quality, particularly micronutrient concentration. Grain Zn and Fe are vital for human health and are often targeted in biofortification breeding [11].

Biofortification is an approach used to improve the nutritional value of the crop and can address insufficient micronutrient content in dietary sources [3]. Genetic biofortification relies on conventional breeding methods and/or modern genetic engineering to increase the nutritional profile in crops [12]. Biofortification through genetic enhancement is recognized as an efficient and cost-efficient strategy for improving nutritional value of food in a sustainable manner to overcome deficiencies in essential minerals [13]. The wheat growing regions encountered with rising temperatures and intermittent rains because of climate change. Due to heat and drought stresses various growth stages of wheat are affected causing reduction in grain assimilate capacity there by decreasing the yield [14]. Temperature and precipitation variations in wheat producing regions will further threaten sustainable access to food and nutrients [15].

Abiotic stresses affect the metal homeostasis which impacts uptake, translocation, and accumulation of Fe and Zn [16]. Studies have reported positive correlations between TGW and micronutrient content, suggesting shared physiological determinants such as source-sink efficiency, grain filling duration, and remobilization of nutrients [11]. However, under heat and drought stress, this relationship can be altered, as stress impacts both biomass accumulation and nutrient translocation. The genetic analysis of polygenic traits aided by the recent advancements in functional genomics and high-throughput DNA sequencing technologies and availability of different DNA chip technology. Recently

association mapping (AM) is employed to unravel the genetic structure of complex traits like grain yield, drought tolerance, and salt stress resilience, host plant tolerance to pathogens phenology, and quality features in bread wheat [17–21].

Genome-wide association studies (GWAS) detect the loci through linkage disequilibrium (LD) analysis. In crops like wheat, which are self-pollinated, LD blocks enhance the power and resolution of mapping, particularly when combined with a diverse mapping panel and high-density SNP markers [22]. In multiple studies GWAS was employed to uncover the genetic basis of iron and zinc content in wheat grains [4,23–25] and QTL mapping strategies [2,26,27], with the goal to detect key genomic regions and putative candidate genes involved in micronutrient regulation.

However, there are limited studies focusing on micronutrient regulation in wheat under abiotic stress conditions [3,28,29]. Although recent studies have begun to explore the genetic architecture of Fe and Zn accumulation under stress conditions, most have been limited to single-stress scenarios or have not integrated yield-related traits like TGW. This underscores the need for comprehensive investigations that account for the combined impact of drought and heat stress on both nutritional quality and grain productivity. By simultaneously analyzing grain Fe content (GFeC), and grain Zn content (GZnC) and thousand-grain weight (TGW), we aim to identify novel and stable MTAs linked to micronutrient accumulation and yield under stress conditions. The findings will provide valuable genetic resources for marker-assisted selection (MAS) and support the development of climate-resilient, nutrient-rich wheat varieties.

## 2. Materials and methodologies

### 2.1. GWAS panel and field evaluations

GWAS was performed on a diverse set of 280 wheat genotypes, including both advanced breeding lines and commercially released cultivars, collected from multiple Indian wheat breeding centers [13]. The panel comprised several stress-tolerant genotypes known for drought and heat resilience. Additionally, high-yielding commercial cultivars like HD3271 and HD3386 were included as check varieties to provide reference points under different conditions. The research was carried out at the ICAR–Indian Agricultural Research Institute (IARI), New Delhi (28° 38′ 30.5″ N, 77° 09′ 58.2″ E, 228 m AMSL) for two seasons 2022−23 and 2023−24 *rabi* (November to March). Weather patterns observed during the wheat growing months (November to March) of both seasons are included in S1 File. The research was undertaken in three conditions namely, Timely sown irrigated (TSIR, as control), Timely sown restricted irrigated (TSRI, for drought), Late sown irrigated (LSIR, for heat) conditions. A total of 5 irrigations was provided for TSRI, i.e., control. In the restricted irrigation (TSRI) treatment, pre-sowing irrigation followed by one irrigation at 21 days after sowing. In the late sown (LSIR) condition, sowing was done in mid-December to expose the crop to high temperatures and five irrigations were provided. An alpha lattice design with two replications was employed for the analysis of the genotypes, and agronomic practices were carried out according to established standards, as indicated in the prior study [30].

For evaluating thousand-grain weight (TGW), grain Fe content (GFeC), and grain Zn content (GZnC), twenty spikes per line were harvested arbitrarily and stored in cloth bags. Hand threshing was done carefully using cloth bags to prevent contamination of metals from the equipment. The GFeC and GZnC levels were determined using approximately twenty grams of grain sample per genotype using a high-throughput, non-destructive ED-XRF spectrometer (energy-dispersive X-ray fluorescence) (model X-Supreme 8000; Oxford Instruments plc, Abingdon, UK), The GFeC and GZnC levels were quantified according to the protocol outlined by Paltridge et al. [31]. The assessment was conducted at ICAR–IIWBR, Karnal (ICAR–Indian Institute of Wheat and Barley Research). The Iron and Zinc concentrations were expressed in mg/kg. TGW was determined by manually counting a randomly selected set of grains from each genotype and an electronic scale was used to determine the weight, and the measurements were recorded in grams.

### 2.2. Phenotypic data analysis

Analysis of variance (ANOVA) of the alpha-lattice design was performed using the "*PBIB*" package in R version 4.3.1. The "*ggplot2*" package in R version 4.3.1 was used for visual representations of phenotypic data through box plots and

histograms. Additionally, "*corr*" package of R version 4.3.1 [32] to calculate the Pearson correlation coefficients under all conditions to evaluate relationships among traits using the BLUP values. The best linear unbiased predictors (BLUPs) for each and combined conditions and heritability also estimated using "*MetaRv6.0 package*" [33]. Broad-sense heritability ($H^2$) was estimated using the formula given below.

$$H^2 = \frac{\sigma^2 g}{(\sigma^2 g + \frac{\sigma^2 \varepsilon}{nRep})}$$

where $\sigma^2 g$ represents the genetic variance component, $\sigma^2 \varepsilon$ denotes the error variance component, and nRep is the number of replicates.

## 2.3. Genotypic analysis, investigation of population structure and LD assessment

CTAB isolation technique [34] was used for DNA extraction from the seedling leaf tissue (7-day-old) under control conditions. DNA concentration was evaluated through gel electrophoresis using agarose gel (0.8%). A total of 268 DNA samples, out of 280 genotypes, were found to meet the required quality standards. These samples were subsequently genotyped using the Axiom Wheat Breeder's Genotyping Array [35] which includes 35,143 SNPs. These SNP markers were filtered based on the following criteria: minor allele frequency (MAF) (less than 5%), missing data (exceeding 20%), and heterozygote frequency (above 25%) with those markers excluded from the analysis. The rest of the 14,625 SNPs, along with the phenotypic data from 268 genotypes, were subjected to additional analysis. The SNP distribution among the chromosomes and the SNP density plot was visualized through the use of the SR-Plots web tool [36].

TASSEL version 5.2.79 was employed Pairwise linkage disequilibrium (LD; $r^2$) values and plotted in relation to genetic distance (in bp) using "*ggplot2*" package in R, as described by Remington et al. [37]. LD decay was determined as the genetic distance at which the $r^2$ value reduced to 50% of its maximum. Through Principal Component Analysis (PCA) using the "*GAPIT* "package version 3.041 population structure was assessed [38]. Further, a phylogenetic tree was generated by implementing Neighbor-Joining (N-J) clustering approach in *TASSEL* 5.2.79 version to construct a phylogenetic tree. A dendrogram was constructed in "*iTOL*" version 6.5.2 [39] by utilizing generated N-J tree file saved in Newick format. Population structure also was generated using the "*STRUCTURE*" software version 2.3.4 [40] and results were visualized using "*iTOL*" version 6.5.2.

## 2.4. Association analysis and In-silco analysis

The BLUP values of GFeC, GZnC and TGW for 268 genotypes, derived from two seasons and three conditions, were employed as phenotypic data in GWAS alongside the corresponding genotypic information. BLUPs account for random effects such as genotype-by-environment interactions, replication effects, and spatial variability. By incorporating mixed models, they reduce the environmental noise and provide phenotypic estimates that more accurately reflect the true genetic potential of each genotype. The identification of significant marker-trait associations (MTAs) was performed using the Bayesian Information and Linkage Disequilibrium Iteratively Nested Keyway (BLINK) model [41] through "*GAPIT*" version 3.041. The expected and observed -log10 (p) values were compared to generate a Q-Q plot, which was used to assess the suitability of the association model. SNPs with p ≤ 0.0001 were considered as significant. A Bonferroni correction was implemented to ensure stringent selection (p = 0.05/overall count of markers). The "*Mg2c*" tool was employed to display the identified SNPs on their corresponding chromosomes [42]. The 100 kb adjacent region of the identified MTAs was used to search for potential candidate genes through the Ensembl Plants data web service, using the IWGSC Reference Sequence v1.0. Sequence v1.0 assembly [43]. By using gene IDs, the proteins and their functions obtained by means of gene annotation taken from the Triticaceae-Gene Tribe website [44]. The Wheat Expression Database was employed to analyze the insilico expression profiles of the identified putative candidate genes [45].

## 2.5. Multi-trait genotype-Ideotype distance index

The Multi-Trait Genotype-Ideotype Distance Index (MGIDI) was used in this study to identify superior genotypes by considering multiple traits. In this study three traits TGW, GFeC and GZnC were considered as highest contributing factors [46] given below.

$$MGIDI_i = \sqrt{\sum_{j=1}^{f} ( F_{ij} - F_j )^2}$$

In this equation, $F_{ij}$ refers to the $i^{th}$ The score of the genotype in the $j^{th}$ factor (i = 1, 2,..., g; j = 1, 2,..., f), where g represents the number of genotypes and f denotes the number of factors, with Fj indicating the score of the ideotype in the $j^{th}$ factor. $MGIDI_i$ represents the genotype-ideotype distance index for the $i^{th}$ genotype. The genotype that closely matched with the ideotype was considered as having lowest MGIDI. R software version 4.2.2 and the 'metan' package was used for MGIDI index calculations.

## 3. Results

### 3.1. Phenotypic data analysis

GFeC, GZnC, and TGW traits indicated a normal frequency distribution (S1 Fig). ANOVA (represented as MSS) demonstrated significant differences between the genotypes for all three traits. The descriptive statistics and heritability components of the association study panel for three traits across various conditions, are displayed in Table 1. The maximum average values for GFeC trait were under the LSIR condition in both the seasons (LSIR_23 - 44.68, LSIR_24 - 45.23). Similarly, GZnC followed the same trend under the LSIR condition, in both years (LSIR_23 - 45.11, LSIR_24 - 47.21). While, the lowest mean values were recorded for GFeC was seen under TSIR condition across the years (TSIR_23 - 37.79, TSIR_24 - 38.06). Likewise, GZnC trait had lowest means under TSIR condition in both the seasons (TSIR_23 - 45.11, TSIR_24 - 45.68). In contrast, TGW trait exhibited an opposite pattern, the lowest mean value was found under the LSIR condition in both seasons (LSIR_23 - 37.9, LSIR_24 - 32.8). TGW trait had the highest mean under the TSIR condition (TSIR_23 - 42.21, TSIR_24 - 40.79) across the years (Fig 1A).

For the trait TGW the CV was ranged from 6.06%to 8.2% during both the seasons. The CV for GFeC ranged from 6.57% to 9.65%, while for GZnC, it varied between 8.92% and 11.38%. The highest heritability was seen for TGW trait (79%) under the LSIR condition during 2024 and the lowest was for GFeC (32%) in 2023 under LSIR condition. The heritability varied from 68% to 79% for TGW, 32% to 62% for GFeC, and 45% to 59% for GZnC suggesting medium to high heritability. To complement the year-wise analysis, pooled two-year means, CV, and heritability estimates for TGW, GFeC, and GZnC are presented in Table 2. Combined ANOVA showed significant genetic variation (p < 0.001) across all traits and conditions. G × E interaction was significant for TGW under all conditions, particularly TSIR and TSRI. GFeC showed minimal G × E and high heritability (0.72) under LSIR, suggesting strong genetic control. GZnC exhibited the highest genetic variance and heritability (0.77) under LSIR, with a notable G × E effect under TSRI, reflecting stress-responsive behavior.

GFeC and GZnC exhibited a strong positive correlation in both seasons in all conditions, as indicated by the Pearson correlation coefficient (p < 0.001) (Fig 1B). The correlation values were as follows TSIR (0.35, 0.12), TSRI (0.23, 0.26), and LSIR (0.19, 0.23). A strong negative correlation was detected between TGW and GZnC under TSIR and TSRI conditions in 2024 (−0.09, −0.19) and LSIR condition in 2023 and 2024 (−0.13, −0.10). Non-significant correlation was noted between TGW and GFeC in both seasons except for TSIR in 2024 where it showed a positive correlation (0.13).

### 3.2. Distribution of SNP markers, population structure analysis and linkage disequilibrium

From the 35K array, a total of 35,143 SNPs were processed for quality filtering, a final dataset of 14,625 high-quality genome-wide SNPs was retained for GWAS analysis. Principal Component Analysis (PCA) (Fig 2A) indicated the

**Table 1. Descriptive Statistics, ANOVA, and Heritability of GFeC, GZnC, and TGW under three conditions of two seasons 2022−23 and 2023−24.**

| Trait | ENV | MSS | Hbs | CV | Mean±SD | Range |
|---|---|---|---|---|---|---|
| **TGW_23** | TSIR | 27.88*** | 0.71 | 6.37 | 42.21±4.41 | 29.5-58.2 |
| | TSRI | 31.77*** | 0.77 | 6.06 | 42.03±4.5 | 29.2-59.3 |
| | LSIR | 27.30*** | 0.79 | 6.16 | 37.9±4.07 | 27.3-53.4 |
| **TGW_24** | TSIR | 33.99*** | 0.74 | 7.16 | 40.79±4.61 | 26.7-54.7 |
| | TSRI | 31.64*** | 0.72 | 7.02 | 41.74±4.49 | 27.6-57.8 |
| | LSIR | 22.90*** | 0.68 | 8.20 | 32.8±3.93 | 20.7-49.1 |
| **GFeC_23** | TSIR | 16.77*** | 0.62 | 6.57 | 37.79±3.41 | 29.8-61.3 |
| | TSRI | 14.87*** | 0.37 | 7.45 | 40.37±3.47 | 30.7-53.6 |
| | LSIR | 23.77*** | 0.32 | 8.91 | 44.68±4.51 | 34.3-72.5 |
| **GFeC_24** | TSIR | 12.06*** | 0.50 | 6.42 | 38.06±3.15 | 27.8-50.1 |
| | TSRI | 33.57*** | 0.51 | 9.65 | 39.55±4.81 | 30.5-77.8 |
| | LSIR | 22.10*** | 0.34 | 8.43 | 45.23±4.28 | 33.8-67.2 |
| **GZnC_23** | TSIR | 42.11*** | 0.59 | 8.92 | 45.11±5.6 | 24.7-86.9 |
| | TSRI | 39.53*** | 0.45 | 9.67 | 47.21±5.68 | 27.5-88.2 |
| | LSIR | 64.87*** | 0.50 | 10.59 | 50.53±7.34 | 28.1-89.6 |
| **GZnC_24** | TSIR | 58.15*** | 0.53 | 11.38 | 45.68±6.56 | 30.4-81.9 |
| | TSRI | 56.38*** | 0.57 | 9.91 | 47.86±6.39 | 32.1-76.6 |
| | LSIR | 66.18*** | 0.50 | 10.07 | 50.6±7.47 | 36.2-86.7 |

ENV-Environment, MSS-Mean sum of square: *** indicates significance at p < 0.001, Hbs-Broad sense heritability, CV-Coefficient of variation, SD-standard deviation.

absence of well-defined sub-populations within the GWAS panel. However, phylogenetic tree-based clustering revealed the presence of eight distinct sub-groups (Fig 2B and 2C). Linkage disequilibrium (LD) was evaluated by calculating the squared correlation coefficient ($r^2$) between SNP pairs was plotted in relation genetic distance (measured in base pairs). The overall LD decay for the entire genome was 4.9 Mb, with the A sub-genome showing a faster decay (3.6 Mb), succeeded by B (5.7 Mb) and D (5.2 Mb) sub-genomes. SNP distribution across the genomes showed 4,477 markers in the A sub-genome, 5,574 in the B sub-genome, and 4,574 in the D sub-genome (S2 Fig). Among individual chromosomes, 4D exhibited the minimum number of polymorphic SNPs (263 SNPs) whereas the maximum SNP count (1,067 SNPs) was on chromosome 1B.

### 3.3. Identification of MTAs for the traits

GWAS was conducted for all the traits using the BLUPs from each treatment across the season, leading to the identification of 43 significant marker trait associations (MTAs) at a threshold P-value of 0.0001(Table 3). However, after applying Bonferroni correction (P-value < 3.42E-6) only 13 associations were considered as significant, while the remaining 30 associations may be classified as suggestive MTAs. A total of 37 MTAs were unique, five MTAs were stable as shown in Manhattan plots (S3 Fig). Additionally, 11 MTAs were detected for combined BLUPS (Fig 3). These MTAs were distributed across 16 chromosomes, as illustrated in Fig 4. The maximum number of MTAs was detected on chromosome 7A, including four associated with GFeC and one with TGW.

Fifteen MTAs were detected under different conditions for TGW On chromosomes 1A, 1B, 1D, 3B, 3D, 4B, 5D, and 7A, and 7B. The $r^2$ values was ranged from 1.09% to 11.17%. A drought specific SNP AX-94512826 (TSRI_23) on chromosome 7A mapped adjacent to *TraesCS7A02G482300* (serine carboxypeptidase-like 51), a proteolysis-related gene

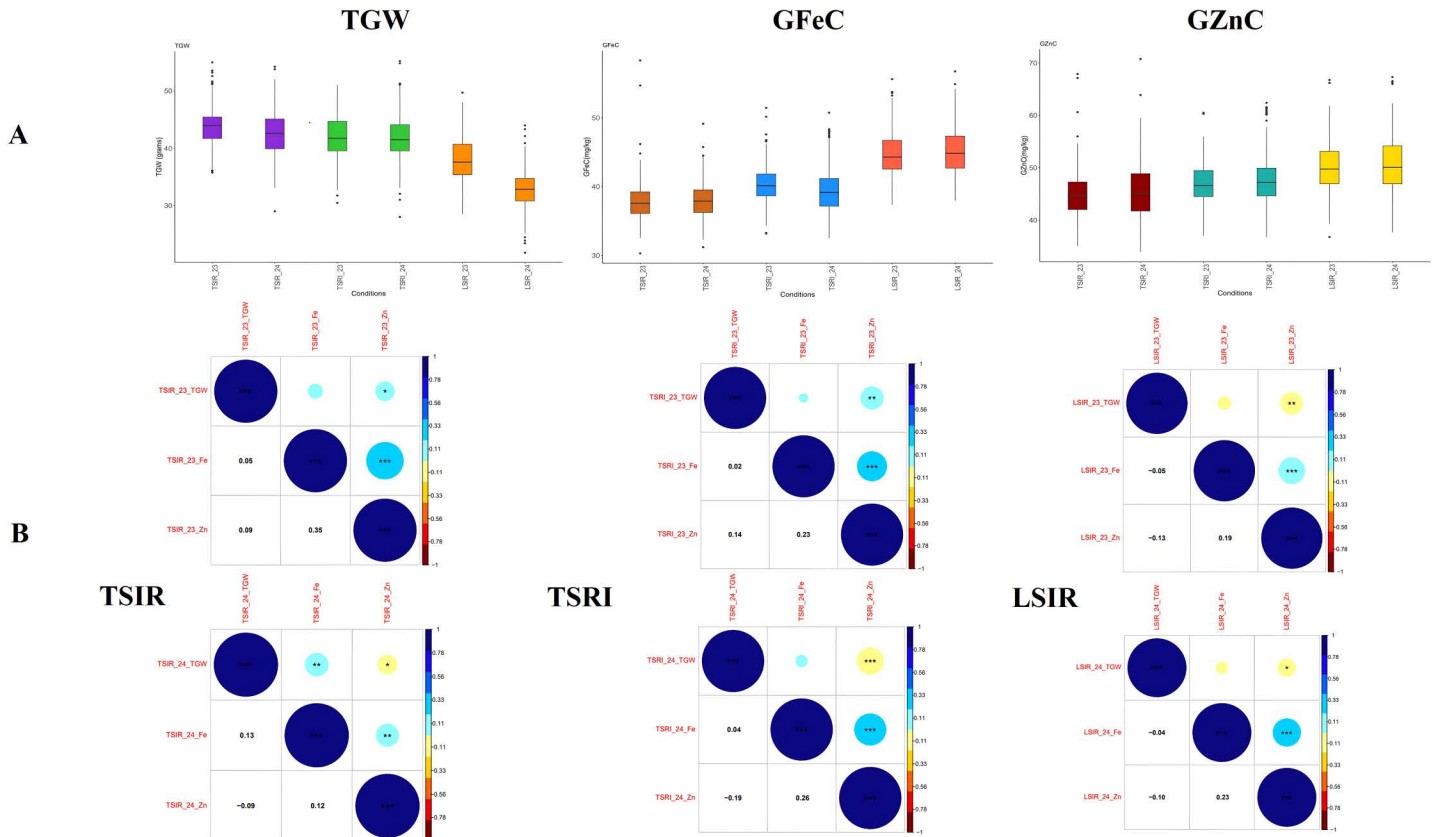

**Fig 1. Phenotypic distribution and trait correlations across the conditions and seasons. (A)** Box Plot Illustrating the Distribution of GFeC, GZnC, and TGW Under three conditions (TSIR, TSRI, LSIR) across the seasons 2022–2023 and 2023–2024 **(B)** Phenotypic correlation coefficients among grain iron content, grain zinc content, and thousand-grain weight.

**Table 2. Summary of pooled performance and variance components for TGW, GZnC, and GFeC under three environments.**

| Trait | ENV | Mean ± SD | CV (%) | Heritability | Genotype Variance | G × Env Variance | Genotype Significance | G × Env Significance |
|---|---|---|---|---|---|---|---|---|
| TGW | TSIR | 41.50 ± 4.51 | 6.76 | 0.5 | 5.08 | 4.55 | $2.62E-08$ *** | $2.40E-08$ *** |
| | TSRI | 41.89 ± 4.50 | 6.54 | 0.43 | 3.9 | 2.93 | $6.06E-06$ *** | 0.00177 ** |
| | LSIR | 35.35 ± 4.00 | 7.18 | 0.66 | 2.98 | 1.13 | $1.20E-17$ *** | $1.49E-05$ *** |
| GZnC | TSIR | 45.40 ± 6.08 | 10.15 | 0.27 | 1.11 | 0.45 | $9.88E-03$ * | 0.454 |
| | TSRI | 47.54 ± 6.04 | 9.79 | 0.31 | 1.13 | 0.19 | $2.60E-03$ ** | 0.712 |
| | LSIR | 50.57 ± 7.41 | 10.33 | 0.72 | 3.81 | 0 | $2.83E-42$ *** | 1 |
| GFeC | TSIR | 37.93 ± 3.28 | 6.5 | 0.5 | 6.61 | 0 | $6.46E-09$ *** | 1 |
| | TSRI | 39.96 ± 4.14 | 8.55 | 0.4 | 5.89 | 6.47 | $3.38E-05$ *** | $1.57E-05$ *** |
| | LSIR | 44.96 ± 4.40 | 8.67 | 0.77 | 11.36 | 0 | $1.37E-52$ *** | 1 |

Note: *$p < 0.05$, **$p < 0.01$, ***$p < 0.001$.

($r^2 = 11.17\%$, $-\log10 (p) = 8.20787$). Also in TSRI_23, AX-95090516 (1B) ($r^2 = 8.93\%$, $-\log10 (p) = 7.16103$) lay close to three tandem genes encoding serine/threonine-protein kinase STN7, phosphoethanolamine N-methyltransferase 1, and a putative clathrin assembly protein, all implicated in protein phosphorylation and stress signaling. Conversely, the SNP

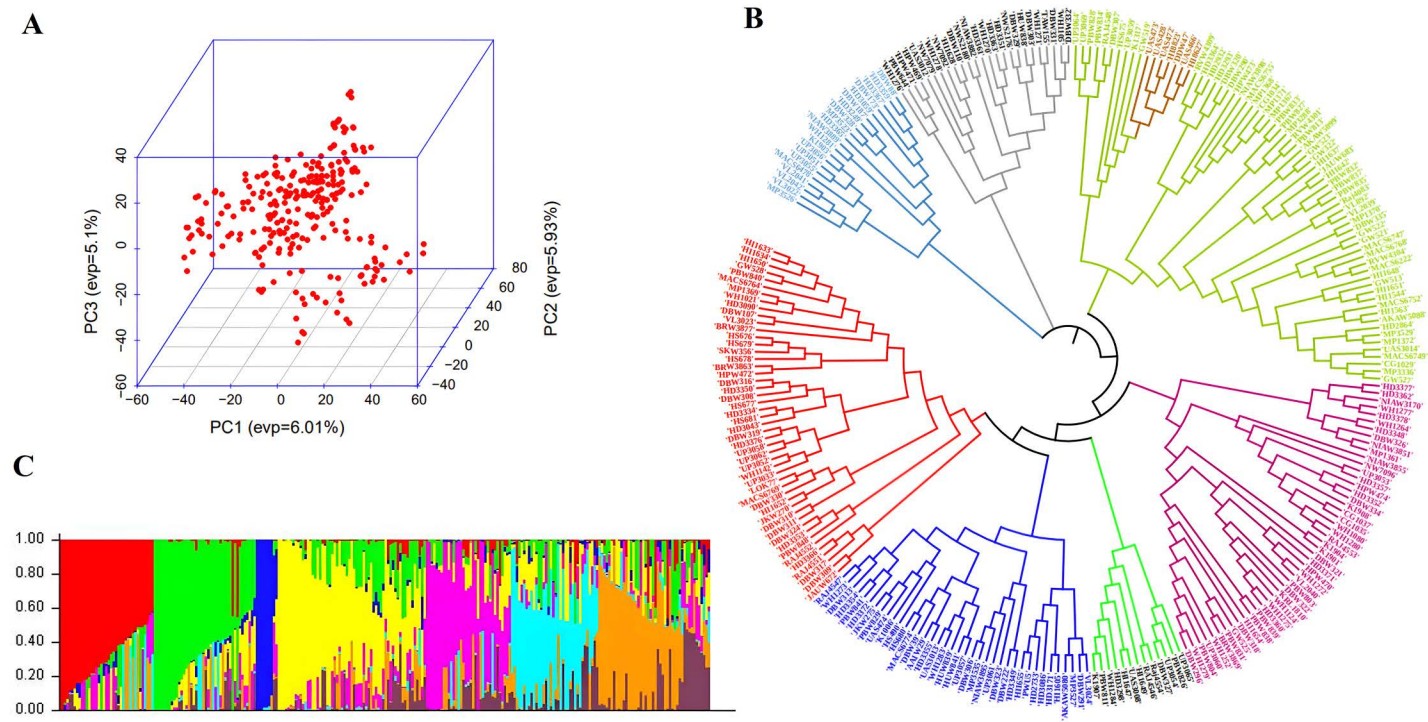

**Fig 2. Population structure of the GWAS panel based on different models. (A)** PCA-Based Clustering (PC1 vs. PC2), **(B)** Neighbor-joining tree using a distance matrix and **(C)** Bar plot illustrating the population structure highlighting the presence of 8 subgroups.

AX-95165363 and AX-94786575 was consistently detected only under control conditions (TSIR and TSIR_CBLUP). SNP AX-95165363 is flanked by *TraesCS1D02G308500* (encoding an LRR protein), *TraesCS1D02G308400* (encoding a calmodulin), and *TraesCS1D02G308200* (encoding an NF1-related kinase regulatory subunit), suggesting a possible role in hormone and calcium-mediated signaling under optimal (non-stress) conditions. SNP AX-94786575 was linked to *TraesCS3B02G310900* and *TraesCS3B02G310800*, encoding a glutamate carboxypeptidase and a LysM receptor-like kinase, respectively. These genes are implicated in plant growth regulation and ABA-mediated stress signaling. Several additional putative candidate genes detected were Probable magnesium transporter, EIN3-binding F-box protein, COP1-interacting protein 7, and Serine/threonine-protein kinase, which are identified as playing a part in stress tolerance (Table 4).

Sixteen marker-trait associations (MTAs) for grain iron concentration (GFeC) were detected across 1A, 1D, 3D, 4A, 4B, 5B, 6D, 7A, and 7D chromosomes. The $r^2$ values for these associations ranged from 0.62% to 8.82%. Among these, the SNP AX-94953068 on chromosome 7A emerged as a stable and consistent signal, detected under both control (TSIR_23, TSIR_CBLUP) and drought conditions (TSRI_CBLUP). This marker is located near *TraesCS7A02G171600*, which encodes galactoside 2-alpha-L-fucosyltransferase, an enzyme implicated in cell wall organization and potentially involved in maintaining iron homeostasis under varying stress conditions. Similarly, AX-94431795 (4B), identified under drought conditions (TSRI_24), is linked to genes associated with protein catabolism, suggesting its role in drought-induced iron remobilization. Heat-specific MTAs were also identified exclusively under the LSIR_23 condition, including AX-94432820 (7A), located near *TraesCS7A02G023300* encoding a RING-H2 finger protein involved in metal ion binding, and AX-95139295 (7A), adjacent to *TraesCS7A02G036200*, encoding chaperonin 60 subunit β1, a protein crucial for refolding stress-denatured proteins. Another significant SNP, AX-95202070 (3D), found under the same condition, lies near

Table 3. Significant MTAs Identified under Different conditions (–log$_{10}$ (p) > 4).

| Trait | Environment | SNP | Chromosome | Position (Mb) | P-value | -log10 (p) | r$^2$ |
|---|---|---|---|---|---|---|---|
| TGW | TSIR_23 | AX-94575939 | 5D | 3.87E+08 | 4.28E-05 | 4.369011 | 9.72 |
| | | **AX-95165363** | 1D | 4.04E+08 | 7.96E-05 | 4.099012 | 4.1 |
| | TSIR_24 | **AX-94786575** | 3B | 4.99E+08 | 3.80E-07 | 6.420379 | 5.77 |
| | | **AX-95165363** | 1D | 4.04E+08 | 1.15E-06 | 5.938136 | 5.82 |
| | | AX-94474486 | 1D | 3.83E+08 | 4.26E-05 | 4.370937 | 1.81 |
| | | AX-94389450 | 1A | 4.81E+08 | 8.88E-05 | 4.051502 | 1.09 |
| | TSIR_CBLUP | **AX-95165363** | 1D | 4.04E+08 | 2.99E-10 | 9.523793 | 6.62 |
| | TSIR_CBLUP | AX-94963644 | 7B | 1256220 | 1.01E-06 | 5.994546 | 5.21 |
| | TSIR_CBLUP | AX-94890423 | 1A | 4.80E+08 | 8.03E-06 | 5.095287 | 2.39 |
| | TSIR_CBLUP | **AX-94786575** | 3B | 4.99E+08 | 7.40E-05 | 4.131007 | 4.87 |
| | TSRI_23 | AX-94512826 | 7A | 6.74E+08 | 6.20E-09 | 8.20787 | 11.17 |
| | | AX-95090516 | 1B | 6.22E+08 | 6.90E-08 | 7.161028 | 8.93 |
| | | AX-94533719 | 4B | 99594802 | 2.67E-07 | 6.57381 | 4.35 |
| | | AX-94590370 | 3D | 4.91E+08 | 4.21E-06 | 5.37552 | 1.8 |
| | TSRI_24 | AX-94424446 | 7B | 5.11E+08 | 7.91E-08 | 7.101559 | 9.37 |
| GFeC | TSIR_23 | AX-94750279 | 7A | 1.27E+08 | 4.27E-05 | 4.369514 | 4.07 |
| | | AX-94755544 | 7D | 1.27E+08 | 5.08E-05 | 4.29422 | 4.95 |
| | | **AX-94953068** | 7A | 1.27E+08 | 5.49E-05 | 4.260481 | 5.94 |
| | TSIR_24 | AX-94694621 | 1A | 4.90E+08 | 1.34E-05 | 4.872536 | 7.04 |
| | | AX-94428309 | 4A | 2.27E+08 | 4.02E-05 | 4.395327 | 7.96 |
| | | AX-94746949 | 1D | 3.94E+08 | 5.01E-05 | 4.299789 | 6.17 |
| | TSIR_CBLUP | **AX-94953068** | 7A | 1.27E+08 | 9.30E-05 | 4.031588 | 4.75 |
| | TSRI_24 | **AX-94431795** | 4B | 5.46E+08 | 1.76E-05 | 4.753913 | 5.97 |
| | TSRI_CBLUP | **AX-94431795** | 4B | 5.46E+08 | 1.14E-07 | 6.941521 | 8.7 |
| | LSIR_23 | AX-95139295 | 7A | 16323054 | 1.11E-05 | 4.954225 | 4.81 |
| | | AX-95202070 | 3D | 38743953 | 8.32E-05 | 4.079669 | 4.29 |
| | | AX-94432820 | 7A | 9372585 | 8.79E-05 | 4.05582 | 3.02 |
| | LSIR_24 | AX-94931570 | 1A | 5.59E+08 | 7.62E-05 | 4.117784 | 1.07 |
| | | AX-94961810 | 1D | 1.60E+08 | 9.20E-05 | 4.036087 | 2.76 |
| | | AX-94453931 | 5B | 5.89E+08 | 9.30E-05 | 4.03146 | 0.63 |
| | LSIR_CBLUP | AX-94689123 | 6D | 1588471 | 9.83E-05 | 4.007234 | 8.83 |
| GZnC | TSIR_23 | **AX-95001849** | 2D | 5.97E+08 | 8.91E-10 | 9.050232 | 12.89 |
| | | AX-94428968 | 4A | 7.36E+08 | 1.77E-07 | 6.752927 | 8.98 |
| | TSIR_24 | AX-94908881 | 7D | 2.80E+08 | 2.43E-05 | 4.613812 | 2.89 |
| | | AX-94513632 | 7D | 24827495 | 7.68E-05 | 4.11467 | 6.47 |
| | | AX-95019083 | 7B | 7.50E+08 | 9.89E-05 | 4.004795 | 6.95 |
| | TSIR_CBLUP | AX-94642295 | 5B | 5.73E+08 | 7.36E-05 | 4.133204 | 1.58 |
| | | AX-95094419 | 7B | 7.21E+08 | 9.64E-05 | 4.016132 | 1.55 |
| | TSRI_23 | **AX-95001849** | 2D | 5.97E+08 | 9.55E-05 | 4.019996 | 10.06 |
| | TSRI_CBLUP | AX-94692056 | 5B | 6.84E+08 | 1.31E-08 | 7.883787 | 3.87 |
| | TSRI_CBLUP | AX-94765773 | 6B | 7.17E+08 | 9.19E-07 | 6.036577 | 4.59 |
| | LSIR_23 | AX-94721306 | 2A | 7.25E+08 | 7.55E-05 | 4.121822 | 2.44 |
| | LSIR_24 | AX-94838752 | 3B | 5.63E+08 | 5.84E-05 | 4.233266 | 3.24 |

MTAs detected multiple times are highlighted in bold text.

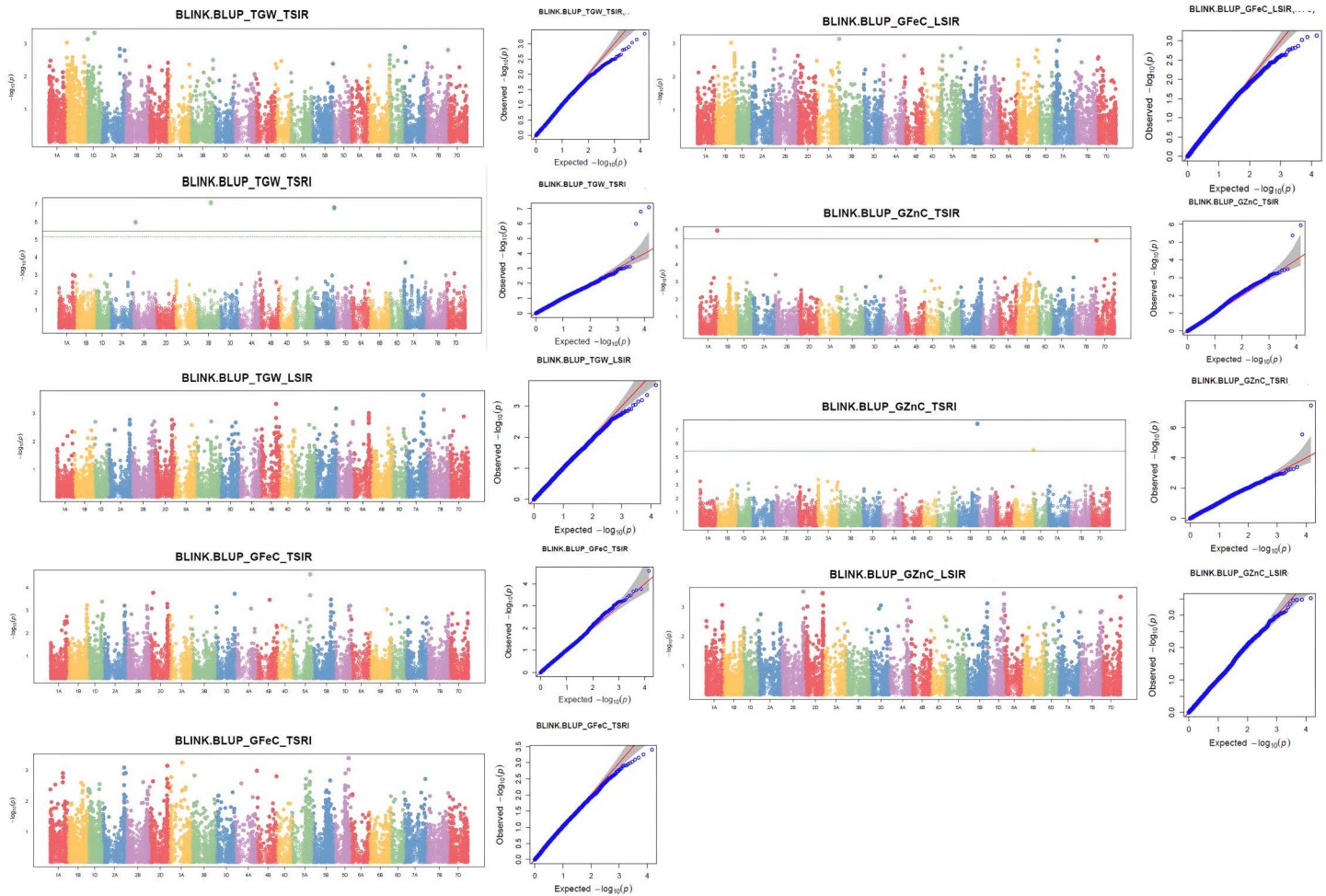

**Fig 3. Manhattan and Q–Q plots displaying significant associations for all traits under TSIR, TSRI, LSIR combined BLUPs.**

*TraesCS3D02G078300*, which encodes a bHLH110 transcription factor, potentially regulating iron uptake and distribution. In addition, AX-94961810 was linked to a gene encoding probable prolyl 4-hydroxylase, a metal ion-binding enzyme likely contributing to iron homeostasis.

For GZnC trait, 12 MTAs were identified on chromosomes 2A, 2B, 2D, 3B, 4A, 5B, 6B, 7B and 7D under different conditions. The $r^2$ value was ranged from 1.54% to 12.89%. The most prominent MTA was AX-95001849 on chromosome 2D (position $5.97 \times 10^8$), which displayed the highest phenotypic variance explained ($r^2 = 12.89\%$, -log10 (p) = 9.05) and was consistently detected under both TSIR_23 and TSRI_23, indicating its stability across normal and restricted irrigation. This SNP is located near *TraesCS2D02G503000*, which encodes a plasma membrane ATPase involved in metal ion transport and homeostasis. Under late sown heat conditions (LSIR_23), the SNP AX-94721306 (2A) ($r^2 = 2.44\%$) was associated with genes such as *TraesCS2A02G492200* and *TraesCS2A02G492100*, encoding a CCCH-type zinc finger protein and a PHD finger-like protein, respectively, both implicated in transcriptional regulation and mRNA splicing, suggesting their role in heat-responsive zinc mobilization. Furthermore, AX-94838752 (3B), detected exclusively under LSIR_24, was located

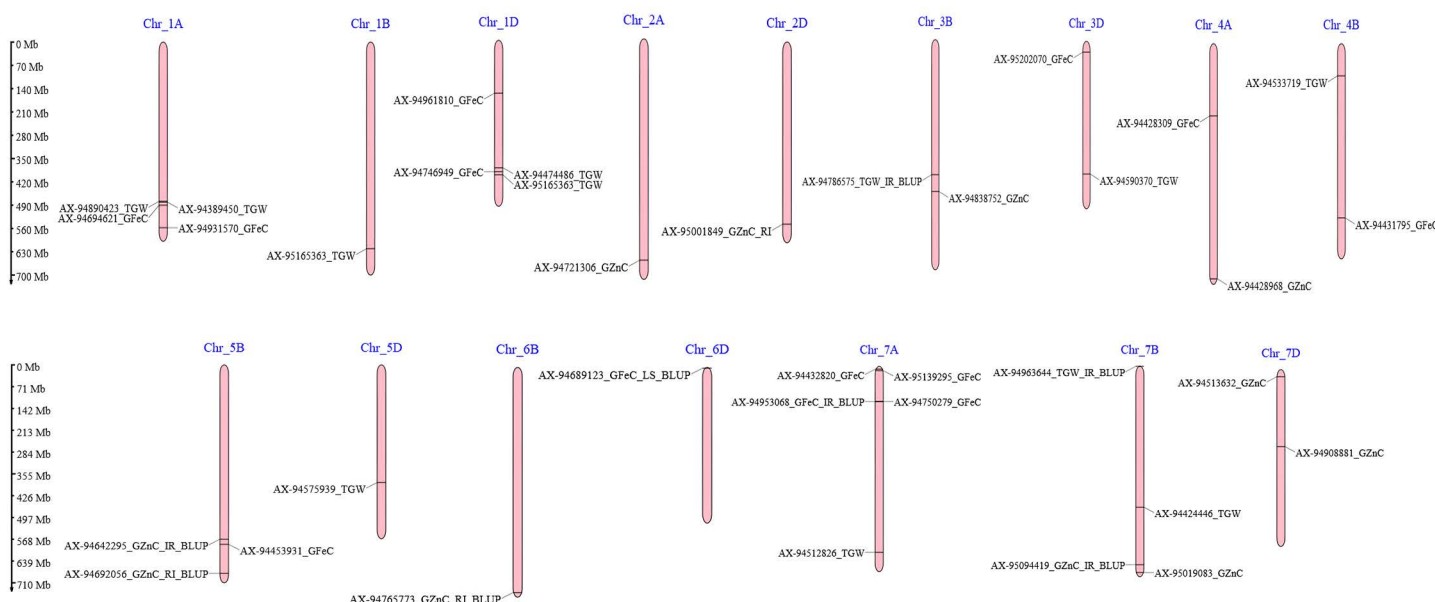

**Fig 4. Identified marker-trait associations (MTAs) for all traits, mapped to their respective chromosomes along with physical positions (in Mb).**

near *TraesCS3B02G353000* (encoding NAD(P)H dehydrogenase, involved in auxin signaling) and *TraesCS3B02G353100* (encoding bHLH112 transcription factor, known for its role in water stress response), indicating a stress-specific regulatory mechanism.

The insilico expression analysis (Fig 5) revealed transcript data such as *TraesCS7A02G036200*, and *TraesCS1D02G132900*, *TraesCS1B02G388700* exhibited higher TPM across all conditions (LSIR, TSIR, and TSRI). Transcripts *TraesCS2D02G503000* and *TraesCS1D02G294400* exhibited higher TPM in both LSIR and TSRI but not in TSIR suggesting a potential involvement of these genes in adaptive responses to abiotic stress.

### 3.4. Selection of superior genotypes through MGIDI

MGIDI index was used to analyse the superior genotypes using 15% of selection intensity to identify top-performing genotypes across multiple traits. This level aligns with standard breeding practices (typically 10–20%) to ensure genetic gain while maintaining diversity [46]. Although no prior simulation was done, this threshold reflects practical field selection pressure in multi-trait studies. Out of 280 genotypes 42 genotypes with low MGIDI values were selected as the best under each condition, as given in S2 File. (Fig 6). The factors influencing MGIDI index were categorized into two groups: higher contributing factors and lower contributing factors. Higher contributing factors in genotype discrimination were located farther from the center of the biplot, while those near the center contributed less. In MGIDI, this spatial pattern reflects the magnitude of each trait's contribution to the multi-trait index. In this study, all three traits (GFeC, GZnC and TGW) were considered as highest contributing factors. In which 9 genotypes were identified as superior across all three conditions for these traits RAJ4546, UP3063, HD3334, DBW296, MP1368, DBW333, UP3058 DBW332 and BRW3863. The pedigree information of these genotypes, along with their corresponding GFeC and GZnC concentrations, is provided in S3 File.

**Table 4. Putative candidate genes within the 100-kb region of the associated marker their protein products and annotations.**

| SNP | Chromosome | TrasID | Protein | Function |
|---|---|---|---|---|
| AX-94575939 | 5D | TraesCS5D02G286500 | Peptidyl-prolyl cis-trans isomerase | secretory vesicle |
| AX-95165363 | 1D | TraesCS1D02G308500 | Leucine-rich repeat protein | hormone-mediated signaling pathway |
| | | TraesCS1D02G308400 | Calmodulin | calcium-mediated signaling |
| | | TraesCS1D02G308200 | NF1-related protein kinase regulatory subunit | ATP binding |
| AX-94786575 | 3B | TraesCS3B02G310900 | Probable glutamate carboxypeptidase | regulation of growth |
| | | TraesCS3B02G310800 | LysM domain receptor-like kinase | positive regulation of abscisic acid-activated signaling pathway |
| AX-94474486 | 1D | TraesCS1D02G284400 | Probable monofunctional riboflavin biosynthesis protein | GTP binding |
| | | TraesCS1D02G284600 | 3-hydroxyacyl-CoA dehydratase | sphingolipid biosynthetic process |
| AX-94389450 | 1A | TraesCS1A02G282200 | Aldo-keto reductase family | oxidoreductase activity |
| | | TraesCS1A02G282400 | NADPH-dependent aldo-keto reductase | response to water deprivation |
| | | TraesCS1A02G282000 | V-type proton ATPase | proton-exporting ATPase activity |
| AX-94512826 | 7A | TraesCS7A02G482300 | Serine carboxypeptidase-like 51 | proteolysis involved in cellular protein catabolic process |
| AX-95090516 | 1B | TraesCS1B02G388900 | Serine/threonine-protein kinase STN7 | protein kinase activity |
| | | TraesCS1B02G388700 | Phosphoethanolamine N-methyltransferase 1 | pollen development |
| | | TraesCS1B02G388800 | Putative clathrin assembly protein | vesicle budding from membrane |
| AX-94590370 | 3D | TraesCS3D02G376800 | Erythroid differentiation-related factor 1 | positive regulation of transcription, DNA-templated |
| | | TraesCS3D02G376700 | Probable magnesium transporter | magnesium ion transport |
| AX-94424446 | 7B | TraesCS7B02G278800 | EIN3-binding F-box protein | ubiquitin-dependent protein catabolic process |
| AX-94963644 | 7B | TraesCS7B02G003000 | COP1-interacting protein 7 | chlorophyll biosynthetic process |
| AX-94890423 | 1A | TraesCS1A02G281100 | BEL1-like homeodomain protein 9 | maintenance of inflorescence meristem identity |
| AX-94750279 | 7A | TraesCS7A02G172600 | 2-hydroxyisoflavanone dehydratase | isoflavonoid biosynthetic process |
| | | TraesCS7A02G172500 | Tuliposide A-converting enzyme | hydrolase activity |
| AX-94755544 | 7D | TraesCS7D02G174600 | Zinc finger protein | zinc ion binding |
| AX-94953068 | 7A | TraesCS7A02G171600 | Galactoside 2-alpha-L-fucosyltransferase | cell wall organization |
| AX-94694621 | 1A | TraesCS1A02G295000 | GDSL esterase | hydrolase activity |
| AX-94428309 | 4A | TraesCS4A02G142300 | Probable serine/threonine-protein kinase | cyclin-dependent protein serine/threonine kinase activity |
| AX-94746949 | 1D | TraesCS1D02G294500 | Probable methyltransferase | |
| | | TraesCS1D02G294400 | Ribonuclease P protein subunit | methyltransferase activity |
| AX-95139295 | 7A | TraesCS7A02G036200 | Chaperonin 60 subunit beta 1 | chaperone cofactor-dependent protein refolding |
| AX-95202070 | 3D | TraesCS3D02G078300 | Transcription factor bHLH110 | DNA-binding transcription factor activity |
| | | TraesCS3D02G078200 | Putative pentatricopeptide repeat-containing protein | |

*(Continued)*

**Table 4.** (Continued)

| SNP | Chromosome | TrasID | Protein | Function |
|-----|-----------|--------|---------|----------|
| AX-94432820 | 7A | TraesCS7A02G023300 | RING-H2 finger protein | metal ion binding |
| AX-94931570 | 1A | TraesCS1A02G391700 | Receptor kinase-like protein | ATP binding |
| | | TraesCS1A02G392100 | Rop guanine nucleotide exchange factor | regulation of pollen tube growth |
| AX-94961810 | 1D | TraesCS1D02G132900 | Probable prolyl 4-hydroxylase | iron ion binding |
| AX-94453931 | 5B | TraesCS5B02G414900 | Succinate dehydrogenase | mitochondrial electron transport |
| AX-94689123 | 6D | TraesCS6D02G002400 | Wall-associated receptor kinase 2 | calcium ion binding |
| AX-95001849 | 2D | TraesCS2D02G503000 | Plasma membrane ATPase | metal ion binding |
| AX-94428968 | 4A | TraesCS4A02G479200 | Probable LRR receptor-like serine/threonine-protein kinase | ATP binding |
| AX-94908881 | 7D | TraesCS7D02G281700 | Mediator of RNA polymerase II transcription subunit 15a | chromatin DNA binding |
| AX-94513632 | 7D | TraesCS7D02G048000 | Aspartyl protease family protein | proteolysis |
| AX-95019083 | 7B | TraesCS7B02G500700 | Putative disease resistance protein | defense response |
| | | TraesCS7B02G500200 | Coiled-coil domain-containing protein | post-mRNA release spliceosomal complex |
| AX-94721306 | 2A | TraesCS2A02G492200 | Zinc finger CCCH domain-containing protein | metal ion binding |
| | | TraesCS2A02G492100 | PHD finger-like domain-containing protein | mRNA splicing |
| AX-94838752 | 3B | TraesCS3B02G353000 | NAD(P)H dehydrogenase | cellular response to auxin stimulus |
| | | TraesCS3B02G353100 | Transcription factor bHLH112 | cellular response to water deprivation |

## 4. Discussion

Heat and drought stress pose major challenges to wheat yield and nutritional quality, highlighting the importance of identifying genetic regions associated with key traits. This study used genome-wide association analysis to uncover loci linked to grain iron, zinc content, and thousand grain weight in a diverse panel evaluated under stress over two years.

ANOVA revealed significant variation for all studied traits among genotypes, suggesting substantial phenotypic diversity within the panel an essential requirement for association mapping [47,48]. As depicted in the boxplots (Fig 1A) it was noted that GFeC and GZnC increased under stress conditions. It emphasizes the adjustment of plants to environmental stresses through various physiological and biochemical mechanisms. Abiotic stresses create variation in the nutrients absorption, their transportation and deposition in the plant system. This results in micronutrient mobilization to grains, as a survival instinct through reproduction during unfavorable conditions [49,50]. This might also be due to the concentration effect, where a reduction in grain yield under stress conditions leads to less dilution of minerals in the grain, thereby increasing the apparent concentration of micronutrients. Several studies have documented this phenomenon, particularly under drought or heat stress, where limited biomass accumulation results in proportionally higher grain zinc and iron concentrations [15,51]. The TGW trait decreased in stress conditions compared to control condition as it is established that that abiotic stresses affect the yield by lowering grain weight and size [52].

The highest range of CV was seen in GZnC followed by GFeC and TGW suggesting the variation in abilities of genotypes to adapt under heat and drought conditions. TGW exhibited the highest heritability among the traits preceded by GZnC which aligns with findings from Devate et al. [28]. The heritability of TGW was highest (68% to 79%) preceded by GZnC (45% to 59%) while GFeC (32% to 62%) had medium to high heritability These findings indicate that the traits are

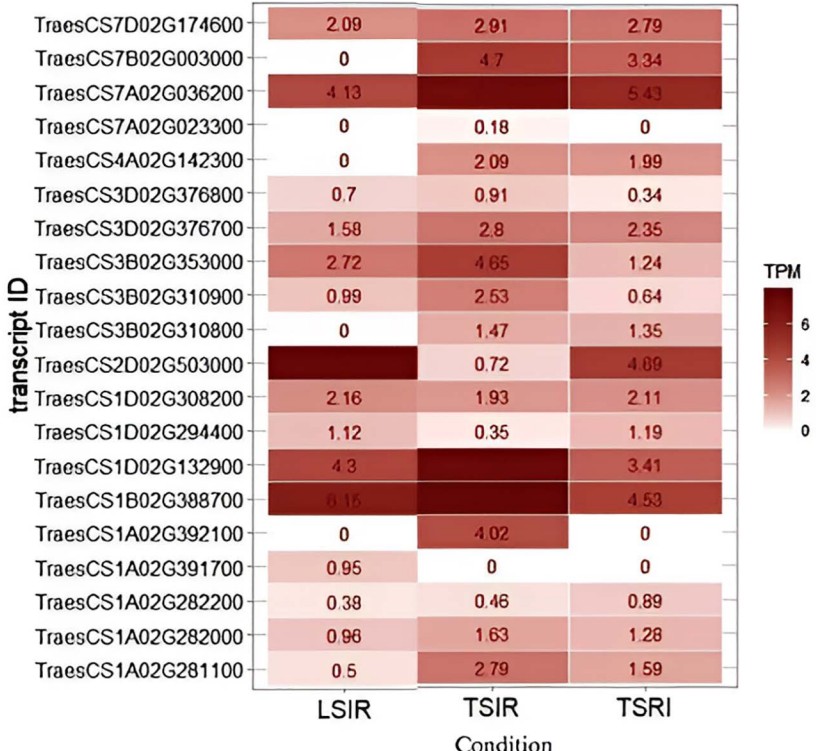

**Fig 5. Heat map showing the expression profiles of candidate genes under control, heat and drought stress conditions.**

primarily controlled by genetic factors, with less environmental influence, allowing them for reliable selection. A similar trend was seen in many earlier studies for GFeC and GZnC [28,53,54]. TGW showed high heritability and strong genotypic effects across the conditions, indicating stable genetic control even under stress, consistent with earlier reports in wheat [55]. In contrast GFeC and GZnC were more sensitive to environmental variation, particularly under drought and heat, as reflected by their moderate heritability and significant environmental effects. Similar trends have been observed in previous studies linking micronutrient accumulation with genotype and environmental interactions [16,51]. A significant positive relationship was found between GFeC and ZnC traits, which aligns with the findings from several studies [3,29]. While, TGW and GZnC traits had a significant negative correlation, except for TSIR and TSRI condition whereas, there was no correlation between TGW and GFeC trait aligned with the findings from prior studies [3,46]. Despite the overall negative correlation between GZnC and TGW, some superior genotypes like UP3063 and BRW3863 showed high values for both traits, indicating potential to overcome this trade-off. Significant positive correlation between GFeC and GZnC suggests a shared genetic or physiological mechanism regulating their accumulation in crops [26,47]. Both micronutrients taken up as divalent cations ($Zn^{2+}$ and $Fe^{2+}$) and often share transporter families like ZIP proteins. They are also chelated by nicotinamide for phloem mobility. These common uptake and transport mechanisms may contribute to their co-accumulation in grains, particularly under stress [55–57]. Significant G×E interactions for TGW and GZnC highlight the role of environmental variability in modulating trait expression, indicating the need for multi-location selection [58]. GZnC showed pronounced sensitivity to drought, suggesting stress-induced effects on zinc uptake and remobilization [15]. In contrast, GFeC showed lower G×E under heat, pointing to greater genetic stability for iron accumulation [51].

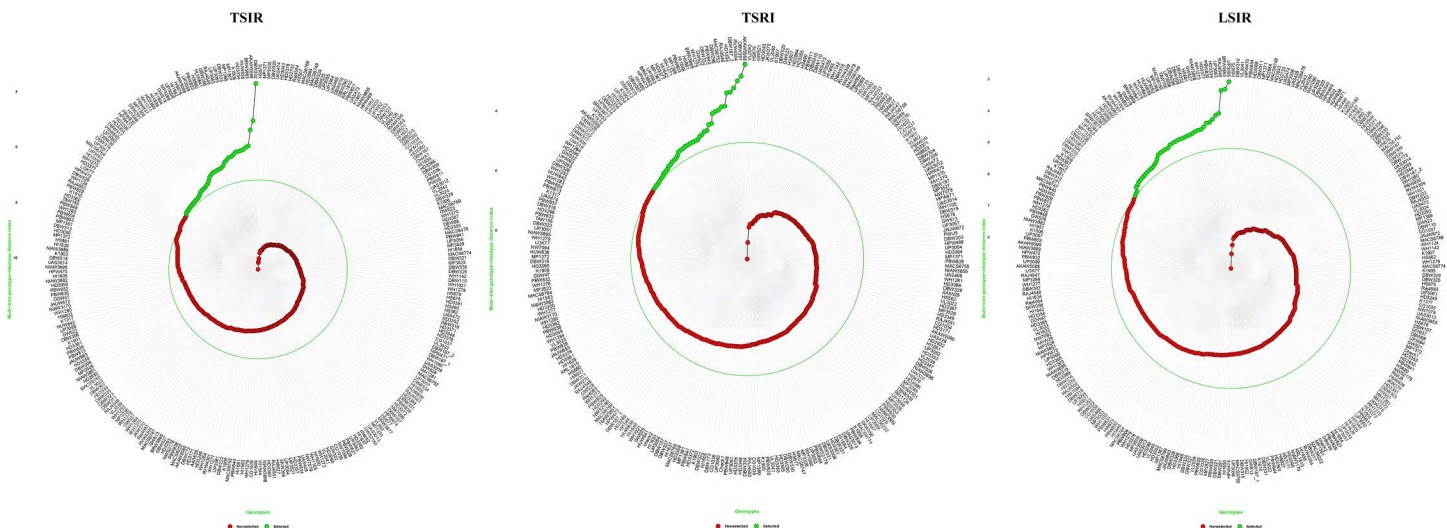

**Fig 6. Selection of superior genotypes using the MGIDI index with a 15% selection intensity.** The central green circle represents the cut-off point defined by the selection intensity, while superior genotypes are indicated by green dots.

Traits that exhibit a positive correlation coupled with additive gene action can be successfully improved simultaneously [12,59]. However, it is imperative to recognize that correlations may vary across various genetic backgrounds and environment conditions [60]. In this study, TGW showed a negative phenotypic correlation with grain Zn content and a slight positive correlation with grain Fe, especially under stress. However, to fully understand the underlying biology, it is critical to assess genetic correlations. For example Velu et al. [15] reported low to moderate genetic correlations between TGW and micronutrients. Similarly, Juliana et al. [61] observed co-localized QTLs for TGW and Zn or Fe. Rathan et al. [4] identified QTLs for TGW and Zn located closely on chromosome 7B, indicating some level of genetic correlation.

Structured populations such as natural collection or landraces further enhance the resolution of mapping due to the considerable genetic variation present and historical recombination. The combination of population structure analysis to minimize false positives and use of robust models boost the power of identification of beneficial genetic loci, thereby accelerating crop improvement [62]. Consequently, PCA was incorporated as a covariate in the GWAS model. PCA based on SNP data revealed a uniform distribution of allelic frequencies with an absence of distinct sub-populations. The association panel used in this research constituted by diverse elite breeding lines selected from various wheat growing zones in India may have resulted in uniform distribution of allelic frequencies. In contrast to PCA, which showed no strong stratification, Neighbor-Joining phylogenetic clustering revealed eight distinct sub-groups. This discrepancy reflects methodological differences: PCA summarizes major variation axes and may miss subtle structure, while tree-based methods like Neighbor-Joining use genetic distances and can detect finer genetic differences [63–65]. Phylogenetic clustering was performed using the Neighbor-Joining method. Although bootstrap support values were not included in the current analysis, future reconstructions using tools such as MEGA X version 10.2.6 [66] or IQ-TREE [67] will incorporate bootstrap replicates to strengthen the confidence in clade groupings.

Accurate LD decay calculation is essential for balancing the choices between marker density and genotyping costs. Marker density in GWAS is determined by how rapidly LD decays across the genome, faster decay necessitates a higher marker density for mapping [68]. A major LD block of size 4.9 cM was measured for the whole genome, as reported by Devate et al. [3]. Among sub genomes, the A sub genome exhibited the most rapid LD decay, preceded by D and the B sub genomes, these results aligned with earlier reports [28,69]. The faster LD decay observed in the A sub-genome may

be attributed to higher historical recombination and the more outcrossing nature of its progenitor, *Triticum urartu*. This genome-specific pattern of LD decay has been consistently reported in wheat, reflecting differences in the evolutionary histories and mating systems of the A, B, and D genome donors [70–72]. Several factors, such as genetic admixtures, population size, genetic drift, mutation, selection pressures, and the mode of pollination, can significantly determine the level of linkage disequilibrium (LD) in plant populations [22,73].

In total, 43 MTAs were detected at a p-value of<0.0001 for both treatment wise and combined BLUPs. After applying the Bonferroni threshold (−log10 (p) > 5.47), only 13 MTAs remained significant. However, MTAs with −log10 (p) > 4 were also considered significant as Bonferroni threshold is too stringent in nature. BLINK model has high statistical power and computational efficiency and minimizes the false positives by using principal components and linked markers as covariates [41]. For polygenic traits, where weaker genetic signals may cumulatively influence variation, it is important to identify the minor/rare alleles [74]. However, such MTAs may be considered as preliminary associations and are suggested for extensive follow-up assessment and validation. The MTAs for TGW were detected on chromosomes 1A,1B,1D,3B,3D,4B,5D,7A,7B. Earlier studies have identified similar MTAs in similar regions using different panels [4,28,75]. The association of stress-responsive SNPs AX-94512826 (7A) and AX-95090516 (1B) with genes involved in proteolysis and phosphorylation highlights key regulatory mechanisms in drought tolerance. These findings underscore the importance of kinase-mediated signaling and protein turnover in maintaining grain development under stress [76,77]. SNP AX-94575939 is found near a candidate gene coding for Cis-trans isomerase of peptidyl-prolyl, playing a role in boosting plant resilience to heat and drought stress through the maintenance of protein stability [78–80]. SNP AX-95165363 was present near the transcribing region that codes leucine-rich repeat (LRR) protein that serves as a key sensor in stress response signaling [81,82]. The BEL1-like homeodomain protein 9 encoded by Putative gene associated with SNP AX-94890423 acts as transcription factor Contributes to growth regulation mechanisms during abiotic stress responses [14].

MTAs related to the GFeC trait were detected distributed across chromosomes 1A, 1D, 3D, 4A, 4B, 5B, 6D, 7A, and 7D these findings align with previous studies [25,26,28,44,61]. Four MTAs (two drought specific) for GFeC trait and one MTA for TGW were detected on chromosome 7A. In agreement with these findings, Tiwari et al. [83] Mapped QTLs linked with GFeC on chromosome 7A further, Rathan et al. [4] detected MTAs for TGW located on the same chromosomal region. Additionally, Leonova et al. [84] identified QTLs for both GFeC and GZnC on 7A. Candidate genes in these regions such as wall-associated kinases, zinc finger proteins, and ABC transporters are likely involved in nutrient uptake and stress response. Examining their domain architectures (e.g., via Pfam) and structural homologs (e.g., via CATH) would clarify whether key functional or catalytic regions are affected by the associated variants. Identification of these regions consistently across the independent studies highlights the significance of 7A chromosome serves as key genetic loci for the integrated improvement of both productivity and quality traits in wheat breeding programs. SNP AX-94432820 identified under heat stress on chromosome 7A was near a gene which encodes for RING-H2 finger protein that binds metal ions and is crucial for responses to abiotic stresses and seed development [85]. The Probable prolyl 4-hydroxylase protein-coding gene was positioned near SNP AX-94961810 (heat specific) involved in iron ion binding and low oxygen response in *Arabidopsis* [86]. The SNP AX-94689123 identified under heat stress, was located near a gene that encodes wall-associated receptor kinase 2 which plays role in calcium ion binding and mediates wide-ranging resistance to fungal diseases [87].

Markers linked to GZnC were identified on chromosomes 2A, 2B, 2D, 3B, 4A, 5B, 6B, 7B, and 7D. Comparable marker-trait associations (MTAs) have been observed on chromosomes 3B, 7B, 2D, 6B, and 7B [61], as well as on chromosome 5B [46,59,88] and chromosome 4A (Devate et al., 2023). Rathan et al. [4], Velu et al. [48], and Crespo-Herrera et al. [88] also detected MTAs for GZnC on chromosome 7B. The aspartyl protease family protein was encoded by a gene found near SNP AX-94513632 that plays a role in protein processing and stress responses [89]. SNP under heat stress, SNP AX-94721306 was near a candidate gene coding for Zinc finger CCCH domain-containing protein involved in metal ion binding hormone signaling, acquisition of immunity against pathogens, and adaptation to stress [90]. Additionally, heat

specific SNP AX-94838752 on chromosome 3B, located near genes encoding NAD(P)H dehydrogenase and the bHLH112 transcription factor, points to the activation of oxidative stress regulation and transcriptional control pathways under abiotic stress. The involvement of bHLH112 in enhancing stress tolerance and regulating metal ion homeostasis, including zinc, has been previously reported [91]. As crop improvement methods necessitates the simultaneous improvement of multiple traits the MGIDI method is a valuable tool. It accelerates the decision making by enabling efficient selection of genotypes [45,92]. In this study, the MGIDI was employed to identify wheat genotypes that performed consistently well for thousand grain weight (TGW), iron (Fe), and zinc (Zn) content across three conditions. Nine genotypes were selected due to their low MGIDI scores, which revealed their desirable balance of agronomic and nutritional attributes. Some of these lines, including UP3063, DBW333, DBW332, and BRW3863, trace their origins to CIMMYT breeding lines, possessing pedigree with high-yielding, stress-resilient parent lines such as MILAN, KAUZ, SOKOLL, and PASTOR [11,58,93]. Lines such as DBW332 and RAJ4546 demonstrated grain zinc concentrations exceeding 60 mg kg$^{-1}$ under stress, aligning with or surpassing levels found in biofortified varieties like 'Zinc Shakti' and select CIMMYT high-Zn lines (~55–65 mg kg$^{-1}$) [15,88]. These genotypes also maintained moderate to high thousand-kernel weights, suggesting that they can have both good grain zinc content and good yield, without the usual trade-off between the two. This suggests the presence of favorable genetic factors supporting the co-enhancement of both micronutrients and yield-related traits. The integration of such diverse and resilient genetic backgrounds emphasizes the effectiveness of MGIDI in selecting genotypes by integrating high yield potential with improved micronutrient content, meeting global objectives for biofortified and climate-resilient wheat cultivars.

## 5. Conclusion

Wheat (*Triticum aestivum* L.) is extensively grown cereal crop and it is an essential source of calories worldwide. Improving grain Fe and Zn concentrations to enhanced stress tolerance through genetic biofortification is a valuable tool to combat the twin challenges of climate amendments and micronutrient malnutrition. In this research, the GWAS has a vital function for discovering genetic factors underlying Fe and Zn concentrations in wheat grains. There was considerable diversity among the evaluated genotypes in TGW, GFeC, and GZnC across all the conditions, indicating the existence of substantial genetic diversity across the panel. A total of 37 unique marker–trait associations (MTAs), including five stable MTAs, were identified. These MTAs were located near candidate genes involved in metal binding and transport mechanisms, highlighting their potential role in micronutrient regulation. SNP AX-94432820 (TraesCS7A02G023300, RING-H2 finger protein) and SNP AX-94953068 (TraesCS7A02G171600, galactoside 2-alpha-L-fucosyltransferase) for GFeC under stress emerged as key candidate genes. MGIDI index facilitated the identification of superior genotypes for TGW, GFeC, and GZnC, which may serve as promising donors in biofortification breeding efforts. This study enhances our knowledge on molecular mechanisms underlying Fe and Zn traits in wheat and also sets the foundation for marker-assisted selection (MAS). MTAs on chromosomes (for GFeC and TGW) and on 7B (for GZnC) represent robust targets for enhancing micronutrient content in wheat. Future research should prioritize fine-mapping key hotspot regions especially on chromosome 7A, which showed stable associations with grain iron concentration and thousand grain weight to pinpoint causal variants. Additionally, functional validation of candidate genes via transcriptomics or gene editing will be crucial to confirm their roles in micronutrient accumulation and yield stability under stress conditions.

## Supporting information

**S1 Fig. Histogram representing the frequency distribution of studied traits across three conditions.**
(TIFF)

**S2 Fig. (A) Linkage disequilibrium (LD) decay across sub genomes (A, B, D) and the entire genome in the GWAS panel (B) SNP density plot depicting the chromosomal distribution of filtered SNPs.**
(TIFF)

**S3 Fig. Manhattan and Q–Q Plots Displaying Significant Associations for TGW, GFeC and GZnC under TSIR, TSRI, and LSIR conditions of two seasons (2022−23 and 2023−24).**
(TIFF)

**S1 File. Weather data recorded across two growing seasons.**
(XLSX)

**S2 File. Multi-trait genotype-ideotype distance index (MGIDI) values for all genotypes across experimental conditions.**
(XLSX)

**S3 File. Pedigree information of selected genotypes identified through MGIDI analysis.**
(XLSX)

## Acknowledgments

Author acknowledges the Indian Council of Agricultural Research Institute (ICAR), New Delhi for Junior Research Fellow Scholarships and the Division of Genetics, ICAR-Indian Agricultural Research Institute, New Delhi, for providing facilities for Ph.D research work.

## Author contributions

**Conceptualization:** Hari Krishna, Neelu Jain, Gyanendra Pratap Singh, Pradeep Kumar Singh.

**Data curation:** Sahana Police Patil, Narayana Bhat Devate, Karthik Kumar Manjunath, P. N. Vinodh Kumar.

**Funding acquisition:** Hari Krishna.

**Investigation:** Hari Krishna, Pradeep Kumar Singh.

**Project administration:** Hari Krishna, Gyanendra Pratap Singh, Pradeep Kumar Singh.

**Resources:** Hanif Khan, Chandra Nath Mishra.

**Supervision:** Neelu Jain, Pradeep Kumar Singh.

**Writing – original draft:** Sahana Police Patil.

**Writing – review & editing:** Sahana Police Patil, Hari Krishna, Narayana Bhat Devate, Karthik Kumar Manjunath, P. N. Vinodh Kumar, Divya Chauhan, Shweta Singh, Chandra Nath Mishra, Neelu Jain, Gyanendra Pratap Singh, Pradeep Kumar Singh.

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
