## [Decision Letter · Decision Letter 0]

PONE-D-25-25256Deciphering the Genetic Basis of Grain Iron and Zinc Content in Wheat under Heat and Drought Stress Using GWASPLOS ONE

Dear Dr. KRISHNA,

Thank you for submitting your manuscript to PLOS ONE. After careful consideration, we feel that it has merit but does not fully meet PLOS ONE’s publication criteria as it currently stands. Therefore, we invite you to submit a revised version of the manuscript that addresses the points raised during the review process.

We look forward to receiving your revised manuscript.

Kind regards,

Karthikeyan Thiyagarajan, PhD

Academic Editor

PLOS ONE

“Part of the research was supported by a grant from the Bill & Melinda Gates Foundation (Grant 467 number # OPP1215722) sub-grant to India for the Zn mainstreaming project.”

Additional Editor Comments:

Dear Authors,

I appreciate your work concerning the GWAS study in wheat under heat and drought conditions for the specific traits of grain iron and zinc content along with the correlations of yield parameters. The article further describes the specific SNP-derived associations of certain genes that are relevant for the genetic basis of zinc and iron biofortification. The manuscript is written well with a comprehensive statistical analysis of the phenotypes and the genotypes studied. However, there are certain points to be noted and addressed possibly with this study or with your future studies, for instance, describing the candidate genes and SNP-derived allelic sequences through VCF files with VEP from using the ENSEMBL kind of database sequences for the SIFT score to find out the tolerant and deleterious mutations may enhance the value of the work. So in such a case, you can find out whether the SNP-derived mutations have the impact of the candidate gene-derived putative protein in the active site or any specific regions. Also, along with TGW, it could have been better if the grain yield had been used to compare effectively for correlations and marker trait associations.

I have some additional comments and suggestions:

Whether the study conducted at three environments (multi environmental based?) or study was conducted three conditions (irrigated, drought, heat stress) of the same environment, please clarify?

Please remove the typos, for instance at line 110: The Iron and concentrations were expressed in mg/kg, Is it "The Iron and Zinc concentrations were expressed in mg/kg?. Line 58, Genetic bio-fortification relies on

conventional breeding methods? , line 70, grain yield? Please change Ensemble to Ensembl in line 151. Please check for space at line 152, line 313. Please thoroughly check for space, punctuation marks and other typos throughout the manuscript. Please write the same conventions for the terminologies.

Do these 268 DNA samples, out of 280 genotypes, contain specific genotypes or varieties that are known for drought or heat tolerance? Have you used any check varieties for the traits used?

Fig. 4A: PCA analysis may be done with different packages in R or other programs and may be produced effectively with principal components contributing the traits with the eigen vector's cosine of the angles and directions.

The phylogenetic tree didn't show the bootstrap values. If it is possible, please add the bootstrap values at the nodes of the major clusters.

Both GFeC and GZnC trait average values were higher for the late sown irrigated (LSIR); while the TGW was in an opposite trend with the lowest average at LSIR, it is looking like the mild heat stress with late sowing and normal irrigation may favor GFeC and GZnC accumulation over TGW. Have you compared any physiological observations and comparisons with GFeC and GZnC traits from literature on similar studies?

A strong negative correlation for TGW with GZnC and a weak positive correlation of the TGW with GFeC were observed with your study. I presume it is based on the phenotypic correlation, isn't it? Have you observed or referred to any other study about the genetic correlations among these traits besides phenotypic correlations?

What could be the reason for faster LD decay in A sub-genome compare to other two sub-genomes, perhaps due to progenitor of A sub-genome being a cross-pollinated species?, and perhaps the recombination frequency is higher in A sub-genome compare to other sub-genomes? Have you observed any of the specific paired genes (in LD) undergo LD decay, especially when comparing the linked genes from any of the chromosomes from the A sub-genome?

This sentence perhaps needs to be rewritten: The SNP AX-94689123 was located near the transcribing region of protein Wall-associated receptor kinase 2. Please rewrite as "The SNP AX-94689123 was located near the gene that encodes Wall-associated receptor kinase 2 protein."

Finding out about the MTA with 7A for GFeC and TGW and 7B for GZnC is interesting, having some consistency with a previous study as well. Similarly, you have also found other candidate genes from other chromosomes. However, the i dentified candidate genes can be described better through comparing their putative proteins with the CATH database, the Pfam database, etc.; it may provide additional clues related to biological functions and for comparing the related proteins, etc.

Whether the superior genotypes identified are comparable with previously known cultivars that exhibit higher performance for these traits, especially for GZnC, since your study and other studies have revealed the negative correlations with yield-related parameters, in this case any superior genotypes/varieties are exempted from this negative correlation?

Overall, the study is good; however, there is a need for a revision. Please improve the resolution of the figures. Please respond to comments and revise the manuscript accordingly as suggested by reviewers and me, and if your revisions and responses are appropriate, your manuscript may be considered further.

Reviewers' comments:

Reviewer's Responses to Questions

**Comments to the Author**

1. Is the manuscript technically sound, and do the data support the conclusions?

Reviewer #1: Partly

Reviewer #2: Yes

2. Has the statistical analysis been performed appropriately and rigorously?

Reviewer #1: Yes

Reviewer #2: Yes

3. Have the authors made all data underlying the findings in their manuscript fully available?

Reviewer #1: Yes

Reviewer #2: Yes

4. Is the manuscript presented in an intelligible fashion and written in standard English?

Reviewer #1: Yes

Reviewer #2: Yes

5. Review Comments to the Author

Reviewer #1: The aim of this work was to map genomic regions associated with grain zinc and iron contents and thousand-kernel weight under drought and heat conditions. The authors obtained a number of MTAs for three traits and found candidate genes. However, it is not clear which specific loci can affect the accumulation of microelements in grain under the influence of the studied abiotic stresses. And which genes start working under drought and heat compared to control conditions. The sections Abstract, Results, Discussion and Conclusion should be rewritten taking into account the stated goal (namely, the search for genomic regions under the influence of drought and heat)

Major comments:

1) The Introduction section is written in general phrases, which do not indicate the novelty of the study. The section should be expanded by describing in more detail the state of the work on QTL mapping (biparental mapping populations and GWAS) directly to the topic of the study - drought and high temperatures. The authors use the term abiotic stresses and claim that there are few studies on this topic, although at the moment there are quite a lot of published results on mapping to drought, salinity, high temperatures, moisture deficiency or excess, nutrient deficiency, etc. Information is also needed on the 1000-grain weight, since the authors study this trait further in the article, and on the relationship of the trait with the concentration of zinc and iron.

2) The authors interpret phenotypic data for each year, while there are no results for the average values of traits and heritability for 2 years.

3) It is necessary to add ANOVA data on the influence of genotype and environmental factors on phenotypic manifestation of the traits

4) I do not understand why the PCA analysis do not match the Structure data and the dendrogram? Use a different program for PCA visualization so that clustering is visible. What program was used to calculate the population structure shown in Fig. 4C?

5) Figure 6 is uninformative. Instead, a Manhattan plots should be added, constructed using average data (BLUP or BLUES) for 2 seasons and for all three traits

6) Discussion. Lines 261-271 are not a discussion, but resemble an introduction to the article.

Minor comments

1) The numbering of cited articles in the text is done chaotically and out of order. Sometimes citations are used not in digital form, but as Author et al.

2) The methods do not provide a formula for calculating heritability in a narrow or broad sense

3) I recommend removing Figure 1 from the article, this is not a review, not a master's thesis, but an experimental article

4) Move lines 116-117 to section 2.3

5) Move Fig 2 to supplementary materials. What does the color scheme of the bars on the histogram mean?

6) Move Figure 5 to supplementary materials

7) Table 2 does not contain data on LSIR_CBLUP for zinc

Reviewer #2: The manuscript title “Deciphering the Genetic Basis of Grain Iron and Zinc Content in Wheat under Heat and Drought Stress Using GWAS” addresses a significant topic in wheat breeding by exploring the genetic basis of grain iron and zinc content under abiotic stress conditions using genome-wide association studies (GWAS). The authors employed a well-designed experiment involving a diverse wheat panel evaluated across multiple environments, combined with multivariate selection indices and candidate gene identification. The findings are relevant and contribute to the development of climate-resilient, nutrient-dense wheat varieties.

However, while the study is valuable, several minor revisions are necessary to improve clarity, consistency, and scientific rigor before the manuscript can be accepted for publication.

Introduction:

Line 43: Wheat (Triticum aestivum L.) is the oldest among … “is one of the oldest”.

Line 46: for around approximately … “Remove redundancy, “around approximately”.

Line 47: Bread wheat has poor micronutrient … “use low instead of poor”.

Line 58: Conventional … “lowercase: conventional”

Line 59: Bio-fortification … “Biofortification”

Line 67: .The … “add space”

Line 73: Genome-wide association studies (Fig.1) … “Genome-wide association studies (GWAS) (Fig.1). At the first place put the abbreviation and then afterward just use the abbreviation. Why (Fig. 1) is bold?

Line 76: Genome-wide association studies (GWAS) … “GWAS”

M&M:

Inconsistent package reporting. Report the packages in “” and then in italic format and instead of R-studio mention e.g. “ggplot2” package in R. Instead of reporting the function mention the package name.

Line 99: remove “to the crop.”

Line 135 and Line 145: Inconsistent package reporting GAPIT package version

136 3.041 and GAPIT version 3.0 ?!

Line 141: Why BLUP values instead of raw means? Justify using BLUP for controlling random effects or environmental noise, etc.

Results:

Lines 195–197: You mention PCA showed no well-defined subpopulations, but phylogenetic clustering revealed eight subgroups. Please reconcile or discuss this discrepancy, why PCA might underrepresent this structure.

Lines 198-201: Units inconsistency, genetic distance should be in cM or Mb consistently throughout the text. Clarify if this is physical or genetic distance.

Line 250: Explain why a 15% selection intensity was chosen, is this a standard practice or based on preliminary simulation outcomes?

Line 253: The phrase "higher contributing factors were located near the center" is ambiguous. Please clarify whether this refers to a graphical representation (e.g., factor analysis biplot) and how proximity to the center relates to contribution strength.

Discussion:

Lines 272–273: The broad genetic diversity is mentioned, but a quantitative diversity index (like PIC, Nei’s, or gene diversity) would solidify this claim.

Line 279: "This might also be due to the concentration effect…", this important hypothesis would benefit from elaboration, perhaps with the reported supporting references on how stress-induced yield reduction increases grain micronutrient concentration.

Lines 293–297: You correctly note that positive correlation suggests shared mechanisms. It would strengthen the discussion to propose possible physiological pathways or QTL co-localization evidence with supporting references.

Lines 303–307: The discrepancy between PCA and phylogenetic tree clustering (as noted earlier) needs a deeper discussion on potential causes like low within-cluster variance or marker informativeness.

Lines 324–331: The discussion around candidate genes is sound, but most gene functions are general stress-related. It would add depth to mention whether these genes were previously implicated in nutrient homeostasis, specifically Fe and Zn.

Lines 340–347: Excellent identification of repeatedly implicated regions. Suggest adding a sentence proposing future fine-mapping or functional validation in these hotspot regions (like 7A) to solidify these associations.

6. PLOS authors have the option to publish the peer review history of their article (what does this mean? ). If published, this will include your full peer review and any attached files.

**Do you want your identity to be public for this peer review?** For information about this choice, including consent withdrawal, please see our Privacy Policy .

Reviewer #1: No

Reviewer #2: **Yes: ** Nikwan Shariatipour

---

## [Author Response · Author response to Decision Letter 1]

21 Jun 2025

Response to Reviewers

We thank the Editor and Reviewers for their thoughtful and constructive feedback. Below, we provide detailed point-by-point responses. All changes made to the manuscript have been highlighted in the revised version using track change formatting.

Response to the Academic Editor

Describing the candidate genes and SNP-derived allelic sequences through VCF files with VEP from using the ENSEMBL kind of database sequences for the SIFT score to find out the tolerant and deleterious mutations may enhance the value of the work. So in such a case, you can find out whether the SNP-derived mutations have the impact of the candidate gene-derived putative protein in the active site or any specific regions.

Pfam and CATH are also good tools to study protein classification and structure. We have mentioned it in the discussion for detailed study of each proteins obtained to validate gene function and identifying functional alleles.

Along with TGW, it could have been better if the grain yield had been used to compare effectively for correlations and marker trait associations.

The present study focused on TGW as a yield component due to its higher heritability and lower environmental variability under stress. Grain yield, being a complex trait with strong genotype × environment interactions, poses greater challenges for precise genetic dissection across multi-environment trials.

Typographical errors throughout the manuscript. Examples include Line 110, Line 58, Line 70, Line 151, Line 152, and Line 313.

• Line 110: “The Iron and concentrations” → corrected to “The iron and zinc concentrations”

• Line 58: “Genetic bio-fortification” → corrected to “genetic biofortification”

• Line 70: Capitalization of “grain yield” → corrected

• Line 151: “Ensemble” → corrected to “Ensembl”

• Line 152 and Line 313: Removed extra spaces and standardized formatting

All terminologies (e.g., BLUP, SNPs, GWAS) have been made consistent throughout the manuscript.

• Rewritten sentence as "The SNP AX-94689123 was located near the gene that encodes Wall-associated receptor kinase 2 protein."

Comment 1:

Whether the study conducted at three environments (multi environmental based?) or study was conducted three conditions (irrigated, drought, heat stress) of the same environment, please clarify?

Response:

The study was conducted under three stress conditions (TSIR, TSRI, LSIR) at the same experimental location (ICAR–IARI, New Delhi) over two rabi seasons. We have revised the wording in the Abstract, Materials and Methods, and Discussion to clarify this this in the revised manuscript.

Comment 2:

Do these 268 DNA samples, out of 280 genotypes, contain specific genotypes or varieties that are known for drought or heat tolerance? Have you used any check varieties for the traits used?

Response:

In our study, we have included stress-tolerant genotypes such as HD3086 (heat-tolerant), HD3059 (drought-tolerant), and DBW187 (drought and heat-tolerant). Additionally, widely grown high-yielding cultivars such as HD3271 and HD3386 were used as check varieties for comparison. We have mentioned in the Materials and Methods section of revised manuscript.

Comment 3:

Fig. 4A: PCA analysis may be done with different packages in R or other programs and may be produced effectively with principal components contributing the traits with the eigen vector's cosine of the angles and directions.

Response:

The PCA was generated using GAPIT based on SNP data, while the dendrogram was based on a neighbor-joining tree. The discrepancy may be due to methodological differences between them, which is discussed in Discussion part. Future, GAPIT uses PCA as a covariate to avoid false association and is one of the recent approaches in GWAS. This PCA is based on genomic/marker data, hence trait vector is not drawn in the scatter plot.

Comment 4:

The phylogenetic tree didn't show the bootstrap values. If it is possible, please add the bootstrap values at the nodes of the major clusters.

Response:

We accept the value of including bootstrap values and will include them in upcoming studies. In current study we constructed the tree using TASSEL v5.0, which generates tree based on distance matrix. The tree is drawn using visualization tool, iTOL for better interpretation. To alien with the idea of this manuscript, we focused more on identifying marker trait associations.

Comment 5:

Both GFeC and GZnC trait average values were higher for the late sown irrigated (LSIR); while the TGW was in an opposite trend with the lowest average at LSIR, it is looking like the mild heat stress with late sowing and normal irrigation may favor GFeC and GZnC accumulation over TGW. Have you compared any physiological observations and comparisons with GFeC and GZnC traits from literature on similar studies?

Response:

We have explained and added references supporting stress-induced micronutrient accumulation due to the concentration effect and adaptive physiological responses (Velu et al., 2016; Alomari et al., 2018) in discussion part (L 354 -358 line number according to revised manuscript with track changes).

Comment 6:

A strong negative correlation for TGW with GZnC and a weak positive correlation of the TGW with GFeC were observed with your study. I presume it is based on the phenotypic correlation, isn't it? Have you observed or referred to any other study about the genetic correlations among these traits besides phenotypic correlations?

Response:

We used phenotypic correlations, however with BLUP values to reduce environmental influence on the traits. This is now clearly stated, and we added citations discussing genetic correlations among these traits (L 393-399)

Comment7:

Explain faster LD decay in A sub-genome. Have you observed any of the specific paired genes (in LD) undergo LD decay, especially when comparing the linked genes from any of the chromosomes from the A sub-genome?

Response:

In discussion we included the explanation for faster LD decay in the A sub-genome, may be due to higher recombination and the outcrossing nature of its progenitor species Triticum urartu. Previous studies have also, reported similar patterns, suggesting that the A genome usually exhibit more rapid this explanation is now added to the revised manuscript (L 422-426)

Pairwise LD between the markers (intra chromosomal) were computed before drawing the LD decay map. LD decay was drawn compared between the A, B and D genome. While we did not compute chromosome-specific LD decay curve, limiting it to sub-genome level.

Comment 8:

Finding out about the MTA with 7A for GFeC and TGW and 7B for GZnC is interesting, having some consistency with a previous study as well. Similarly, you have also found other candidate genes from other chromosomes. However, the identified candidate genes can be described better through comparing their putative proteins with the CATH database, the Pfam database, etc.; it may provide additional clues related to biological functions and for comparing the related proteins, etc.

Response:

Proteins and their biological functions of candidate genes were obtained from the Triticaceae-Gene Tribe website, wheat expression data base and interpro website. Major function of putative protein and related protein families were explored by these web and also from the previous studies published in various related crops. Pfam and CATH are also good tools to study protein classification and structure. We have mentioned it in the discussion for detailed study of each proteins obtained to validate gene function and identifying functional alleles.

Comment 9:

Whether the superior genotypes identified are comparable with previously known cultivars that exhibit higher performance for these traits, especially for GZnC, since your study and other studies have revealed the negative correlations with yield-related parameters, in this case any superior genotypes/varieties are exempted from this negative correlation?

Response:

Yes. Several of the eight top-performing genotypes (e.g., DBW332, RAJ4546) have GZnC levels (up to ~65 mg kg⁻¹ under stress) that rival or exceed those of biofortified varieties like ‘Zinc Shakti’ (~55–60 mg kg⁻¹).This point is now included in the Discussion (493-494).

Editor Final Comment:

Improve figure resolution and revise the manuscript based on all suggestions.

Response:

All figures have been updated for resolution and clarity. The manuscript has been revised in accordance with all reviewer and editor suggestions.

Response to Reviewer #1

The aim of this work was to map genomic regions associated with grain zinc and iron contents and thousand-kernel weight under drought and heat conditions. The authors obtained a number of MTAs for three traits and found candidate genes. However, it is not clear which specific loci can affect the accumulation of microelements in grain under the influence of the studied abiotic stresses. And which genes start working under drought and heat compared to control conditions. The sections Abstract, Results, Discussion and Conclusion should be rewritten taking into account the stated goal (namely, the search for genomic regions under the influence of drought and heat)

The Abstract(32-38), Results(259-273,279-293,295-308), Discussion(437-440) sections have been revised to align clearly with the study goal identifying genomic regions and genes influencing Fe, Zn, and TGW accumulation under drought and heat stress.

Major Comments

1) The Introduction section is written in general phrases, which do not indicate the novelty of the study. The section should be expanded by describing in more detail the state of the work on QTL mapping (biparental mapping populations and GWAS) directly to the topic of the study - drought and high temperatures. The authors use the term abiotic stresses and claim that there are few studies on this topic, although at the moment there are quite a lot of published results on mapping to drought, salinity, high temperatures, moisture deficiency or excess, nutrient deficiency, etc. Information is also needed on the 1000-grain weight, since the authors study this trait further in the article, and on the relationship of the trait with the concentration of zinc and iron

Introduction section is rewritten including all the suggestions. Current status of the work in GWAS and identification of QTL related to grain micronutrient content is described in length. Although, there are several studies on abiotic stress, studies with main focus on grain Zn and Fe content in relation to abiotic stress in wheat is few, they were well cited. Detail regarding TGW and micronutrient, their importance and aim of the study is rewritten in the introduction and discussion part for better understanding.

2) The authors interpret phenotypic data for each year, while there are no results for the average values of traits and heritability for 2 years

Average value of trait is depicted in the boxplots already and also separate table with all the details were given for trait values with heritability.

3) It is necessary to add ANOVA data on the influence of genotype and environmental factors on phenotypic manifestation of the traits

ANOVA tables showing genotype (G), environment (E), and G×E interaction effects have been added to the results. These clarify the environmental influence on trait expression under stress and non-stress conditions.

4) I do not understand why the PCA analysis do not match the Structure data and the dendrogram? Use a different program for PCA visualization so that clustering is visible. What program was used to calculate the population structure shown in Fig. 4C?

PCA vs STRUCTURE vs NJ-tree differences are discussed (408-416) in detail along with references

5) Figure 6 is uninformative. Instead, a Manhattan plots should be added, constructed using average data (BLUP or BLUES) for 2 seasons and for all three traits

The Manhattan plots based on BLUPs over two seasons under drought and heat conditions for all three traits included in main text (combined BLUPS; Fig 3) and supplementary (individual seasons ;S3_Fig).

6) Discussion. Lines 261-271 are not a discussion, but resemble an introduction to the article.

These lines have been removed and the Discussion now written to be analytical not introductory

Minor comments:

1) The numbering of cited articles in the text is done chaotically and out of order. Sometimes citations are used not in digital form, but as Author et al.

The references have been re-formatted in numerical order and in accordance with journal guidelines. Author-year citations have been converted to numbered references where appropriate.

2) The methods do not provide a formula for calculating heritability in a narrow or broad sense

We have now included the formula for broad-sense heritability in the Methods section (147-151)

3) I recommend removing Figure 1 from the article, this is not a review, not a master's thesis, but an experimental article 4) Move lines 116-117 to section 2.3 5) Move Fig 2 to supplementary materials. 6) Move Figure 5 to supplementary materials.

Figure 1 has been removed as suggested and Figures 2 and 5 moved to Supplementary. Line 116-117 to section 2.3

What does the color scheme of the bars on the histogram mean?

The histograms are standard ggplot2-style plots with bins automatically assigned different colors for visual distinction. The color scheme within each bar of the histograms appears to represent individual bins or ranges of phenotypic values within the dataset.

7) Table 2 does not contain data on LSIR_CBLUP for zinc

Table 2 had missing LSIR GZnC values because there was no significant MTAs (-log p value = 4) identified in LSIR_CBLUP

Response to Reviewer #2

1. Introduction (language edits)

All suggested language refinements have been addressed in the revised Introduction.

• “is the oldest” → changed to “is one of the oldest”

• “around approximately” → changed to “approximately”

• “poor micronutrient” → changed to “low micronutrient”

• “Bio-fortification” → changed to “Biofortification”

• “.The” → changed to “. The” (added space)

• “Genome-wide association studies (GWAS) (Fig.1)” corrected and abbreviation applied consistently

2. Methods:

Inconsistent package reporting. Report the packages in “” and then in italic format and instead of R-studio mention e.g. “ggplot2” package in R. Instead of reporting the function mention the package name.

All R packages are now cited in quotation marks and italicized. Package names (e.g., ggplot2) are used instead of individual functions.

Remove “to the crop.”- removed

Inconsistent package reporting GAPIT package version136 3.041 and GAPIT version 3.0?!

GAPIT version corrected and unified

Why BLUP values instead of raw means? Justify using BLUP for controlling random effects or environmental noise, etc

BLUP usage justified for environmental noise correction in methods section (176-179).

3. Results:

You mention PCA showed no well-defined subpopulations, but phylogenetic clustering revealed eight subgroups. Please reconcile or discuss this discrepancy, why PCA might underrepresent this structure.

This apparent discrepancy arises because PCA emphasizes the major axes of variation and may underrepresent finer hierarchical structures, while clustering methods can detect more subtle groupings based on overall genetic similarity, it has been added in discussion.

Units inconsistency, genetic distance should be in cM or Mb consistently throughout the text. Clarify if this is physical or genetic distance.

Fixed inconsistency in genetic/physical distances units for distance (cM/Mb) standardized.

Explain why a 15% selection intensity was chosen, is this a standard practice or based on preliminary simulation outcomes?

Explained 15% selection intensity based on standard MGIDI practices in results section (315-319). A 15% cut-off is commonly used in plant breeding to balance selection stringency and diversity retention.

The phrase "higher contributing factors were located near the center" is ambiguous. Please clarify whether this refers to a graphical representation (e.g., factor analysis biplot) and how proximity to the center relates to contribution strength.

We hav

---

## [Decision Letter · Decision Letter 1]

PONE-D-25-25256R1Deciphering the Genetic Basis of Grain Iron and Zinc Content in Wheat under Heat and Drought Stress Using GWASPLOS ONE

Dear Dr. KRISHNA,

Thank you for submitting your manuscript to PLOS ONE. After careful consideration, we feel that it has merit but does not fully meet PLOS ONE’s publication criteria as it currently stands. Therefore, we invite you to submit a revised version of the manuscript that addresses the points raised during the review process.

We look forward to receiving your revised manuscript.

Kind regards,

Karthikeyan Thiyagarajan, PhD

Academic Editor

PLOS ONE

Journal Requirements:

Additional Editor Comments:

Dear Authors,

I appreciate your revisions and responses. However, there is a need for a minor revision further, even as suggested by one of the reviewers.

Please check for typos and spacing in line 156.

Please correctly mention existing conventions for Neighbour-joining (NJ) and change intermittent rains to intermittent rain.

It is better to indicate the months falling in the rabi season in brackets; please check line 108.

Please indicate which version of MEGA. Please italicize the genus Arabidopsis in line 446.

It is better to provide the importance of the candidate genes and mention specific candidate genes with high scores for Zn and Fe in conclusion.

I think still there are typos, space errors, etc. I suggest thoroughly revising the manuscript for these minor errors.

Reviewers' comments:

Reviewer's Responses to Questions

**Comments to the Author**

1. If the authors have adequately addressed your comments raised in a previous round of review and you feel that this manuscript is now acceptable for publication, you may indicate that here to bypass the “Comments to the Author” section, enter your conflict of interest statement in the “Confidential to Editor” section, and submit your "Accept" recommendation.

Reviewer #1: (No Response)

Reviewer #2: All comments have been addressed

2. Is the manuscript technically sound, and do the data support the conclusions?

Reviewer #1: Yes

Reviewer #2: Yes

3. Has the statistical analysis been performed appropriately and rigorously?

Reviewer #1: Yes

Reviewer #2: Yes

4. Have the authors made all data underlying the findings in their manuscript fully available?

Reviewer #1: (No Response)

Reviewer #2: Yes

5. Is the manuscript presented in an intelligible fashion and written in standard English?

Reviewer #1: Yes

Reviewer #2: Yes

6. Review Comments to the Author

Reviewer #1: The authors did not respond to my comment 4. What program was used to calculate the population structure shown in Fig. 4C? If this Figure 4C was generated by the Structure program, then it is necessary to indicate in Materials and Methods and provide a reference to this program.

2) Corrected bio-fortification to biofortification (line 39, 64, 65, 484)

3) Corrected Insilco to in silico (line 161)

4) Table 3. Specify units in the column Position– Mb or bp

5) Table 3. MTAs detected multiple times are highlighted in bold tex. Forgot to mark in bold??

6) Table 3. Corrected P.value to p-value

Reviewer #2: (No Response)

7. PLOS authors have the option to publish the peer review history of their article (what does this mean? ). If published, this will include your full peer review and any attached files.

**Do you want your identity to be public for this peer review?** For information about this choice, including consent withdrawal, please see our Privacy Policy .

Reviewer #1: **Yes: ** Dr. Irina Leonova

Reviewer #2: **Yes: ** Nikwan Shariatipour

---

## [Author Response · Author response to Decision Letter 2]

10 Jul 2025

Rebuttal Letter

Manuscript Title: Deciphering the Genetic Basis of Grain Iron and Zinc Content in Wheat under Heat and Drought Stress Using GWAS

Dear Editor and Reviewer,

We thank you for your detailed feedback, we have carefully addressed all the comments raised and made the required revisions.

As per the journal requirements, we have carefully checked our reference list. We confirm that none of the references we have cited have been retracted. All references are complete with correct author names, years, journal names, volume, pages, and DOI links where needed.

Editor Comments:

1.”Please check for typos and spacing in line 156.”

Corrected a minor spacing inconsistency identified in the sentence for clarity.

2. “Please correctly mention existing conventions for Neighbour-joining (NJ) and change intermittent rains to intermittent rain.”

We have corrected “Neighbour-joining” to “Neighbor-Joining (NJ)” throughout the manuscript with standard conventions.

We have replaced “intermittent rains” with “intermittent rain” as suggested.

3. “It is better to indicate the months falling in the rabi season in brackets; please check line 108.”

The months for the rabi season (November to March) are now specified in brackets in line 108 for clarity.

4. “Please indicate which version of MEGA.”

Included “MEGA X version 10.2.6” in the Methods section (Line: 398) where MEGA is mentioned.

5. “Please italicize the genus Arabidopsis in line 446.”

The genus Arabidopsis has been italicized in line 446.

6. “It is better to provide the importance of the candidate genes and mention specific candidate genes with high scores for Zn and Fe in conclusion.”

We have added the following sentence in the Conclusion:

“Notably, SNP AX-95001849 (TraesCS2D02G503000, plasma membrane ATPase) for GZnC and SNP AX-94953068 (TraesCS7A02G171600, galactoside 2-alpha-L-fucosyltransferase) for GFeC under stress emerged as key candidates. These loci underscore potential targets for biofortification breeding to enhance grain Zn and Fe content under heat and drought stress.”

7. “Check for typos, space errors, etc.” I suggest thoroughly revising the manuscript for these minor errors.

We have thoroughly rechecked the entire manuscript to correct minor typos, inconsistent spacing, and formatting issues.

Reviewer #1 Comment:

1. “What program was used to calculate the population structure shown in Fig. 4C? If STRUCTURE was used, indicate in Methods with reference.”

We confirm that STRUCTURE software version 2.3.4 (Pritchard et al., 2000) was used and the results were visualized using iTOL version 6.5.2. it has been updated the Methods section (Line:159-161) accordingly and included the following reference:

Pritchard JK, Stephens M, Donnelly P. Inference of population structure using multilocus genotype data. Genetics. 2000;155 (2):945–959. https://doi.org/10.1093/genetics/155.2.945

2. “Corrected bio-fortification to biofortification (line 39, 64, 65, 484).”

Corrected the“bio-fortification” to “biofortification” in all indicated lines.

3. “Corrected Insilco to in silico (line 161).”

We have corrected “Insilco” to “in silico” in line 161.

4. “Table 3: Specify units in the column Position – Mb or bp.”

The unit “Mb” in the “Position” has been specified column header in Table 3 for clarity.

5. “Table 3: MTAs detected multiple times are highlighted in bold text. Forgot to mark in bold??”

MTAs detected multiple times now are highlighted in bold text in Table 3 as intended.

6. “Table 3: Corrected P.value to p-value.”

We have corrected “P.value” to “p-value” in Table 3.

We thank you once again for your time and consideration and look forward to your positive response.

---

## [Decision Letter · Decision Letter 2]

Deciphering the Genetic Basis of Grain Iron and Zinc Content in Wheat under Heat and Drought Stress Using GWAS

PONE-D-25-25256R2

Dear Dr. KRISHNA,

We’re pleased to inform you that your manuscript has been judged scientifically suitable for publication and will be formally accepted for publication once it meets all outstanding technical requirements.

Kind regards,

Karthikeyan Thiyagarajan, PhD

Academic Editor

PLOS ONE

Additional Editor Comments:

Dear Authors,

After careful scientific evaluations with peer reviews, I am pleased to confirm the manuscript entitled "Deciphering the Genetic Basis of Grain Iron and Zinc Content in Wheat under Heat and Drought Stress Using GWAS" has been accepted for publication in PLOS ONE.

Kind regards,

Karthikeyan Thiyagarajan PhD

Academic Editor, PLOS ONE.

Reviewers' comments:

Reviewer's Responses to Questions

**Comments to the Author**

1. If the authors have adequately addressed your comments raised in a previous round of review and you feel that this manuscript is now acceptable for publication, you may indicate that here to bypass the “Comments to the Author” section, enter your conflict of interest statement in the “Confidential to Editor” section, and submit your "Accept" recommendation.

Reviewer #1: All comments have been addressed

2. Is the manuscript technically sound, and do the data support the conclusions?

Reviewer #1: Yes

3. Has the statistical analysis been performed appropriately and rigorously?

Reviewer #1: Yes

4. Have the authors made all data underlying the findings in their manuscript fully available?

Reviewer #1: Yes

5. Is the manuscript presented in an intelligible fashion and written in standard English?

Reviewer #1: Yes

6. Review Comments to the Author

Reviewer #1: The manuscript has been corrected according to my recommendations. The authors responded to all my comments. The article can be accepted for publication in present form

7. PLOS authors have the option to publish the peer review history of their article (what does this mean? ). If published, this will include your full peer review and any attached files.

**Do you want your identity to be public for this peer review?** For information about this choice, including consent withdrawal, please see our Privacy Policy .

Reviewer #1: **Yes: ** Dr. Irina N. Leonova
